# Mitochondrial respiration contributes to the interferon gamma response in antigen-presenting cells

Michael C Kiritsy[1], Katelyn McCann[1,2], Daniel Mott[1], Steven M Holland[2], Samuel M Behar[1], Christopher M Sassetti[1]*, Andrew J Olive[3]*

[1]Department of Microbiology and Physiological Systems, University of Massachusetts Medical School, Worcester, United States; [2]Immunopathogenesis Section, Laboratory of Clinical Immunology and Microbiology, National Institute of Allergy and Infectious Diseases, National Institutes of Health, Bethesda, United States; [3]Department of Microbiology & Molecular Genetics, College of Osteopathic Medicine, Michigan State University, East Lansing, United States

*For correspondence:
christopher.sassetti@umassmed.edu (CMS);
oliveand@msu.edu (AJO)

Competing interest: The authors declare that no competing interests exist.

**Abstract** The immunological synapse allows antigen-presenting cells (APCs) to convey a wide array of functionally distinct signals to T cells, which ultimately shape the immune response. The relative effect of stimulatory and inhibitory signals is influenced by the activation state of the APC, which is determined by an interplay between signal transduction and metabolic pathways. While pathways downstream of toll-like receptors rely on glycolytic metabolism for the proper expression of inflammatory mediators, little is known about the metabolic dependencies of other critical signals such as interferon gamma (IFNγ). Using CRISPR-Cas9, we performed a series of genome-wide knockout screens in murine macrophages to identify the regulators of IFNγ-inducible T cell stimulatory or inhibitory proteins MHCII, CD40, and PD-L1. Our multiscreen approach enabled us to identify novel pathways that preferentially control functionally distinct proteins. Further integration of these screening data implicated complex I of the mitochondrial respiratory chain in the expression of all three markers, and by extension the IFNγ signaling pathway. We report that the IFNγ response requires mitochondrial respiration, and APCs are unable to activate T cells upon genetic or chemical inhibition of complex I. These findings suggest a dichotomous metabolic dependency between IFNγ and toll-like receptor signaling, implicating mitochondrial function as a fulcrum of innate immunity.

## Editor's evaluation

In this article, Olive and colleagues used a genetic screen to identify complex I (CI) of the electron transport chain (ETC) as a regulator of IFNγ-mediated gene expression in macrophages. They attribute this role of CI to effects on the activity of the JAK-STAT pathway downstream of the IFNγ receptor. That CI (or perhaps ETC) activity can acutely regulate JAK-STAT signaling has interesting implications for the metabolic regulation of signal transduction, and the underpinning basis would be important to elucidate in future studies.

## Introduction

During the initiation of an adaptive immune response, the antigen-presenting cell (APC) serves as a point of integration where tissue-derived signals are conveyed to T cells. Myeloid APCs, such as macrophages and dendritic cells (DCs), are responsible for the display of specific peptides in complex with major histocompatibility complex (MHC) molecules, and for the expression of co-signaling factors

that tune the T cell response (*Sharpe, 2009*). The expression of stimulatory or inhibitory co-signaling molecules depends on the local immune environment and activation state of the APC (*Attanasio and Wherry, 2016*). In particular, interferon gamma (IFNγ) stimulates the surface expression of MHC proteins (*Ting and Trowsdale, 2002*; *Buxadé et al., 2018*; *Herrero et al., 2001*; *Reith et al., 2005*; *Rock et al., 2016*; *Steimle et al., 1994*; *Wheelock, 1965*), co-stimulatory proteins such as CD40, and the secretion of cytokines like IL-12 and IL-18 (*Tominaga et al., 2000*), to promote T cell activation and the production of IFNγ-producing T-helper type 1 (Th1) effector cells (*O'Shea and Paul, 2002*; *Schneider et al., 2010*; *Trinchieri, 2003*; *Johnson-Léger et al., 1997*; *Alderson et al., 1993*). In the context of local inflammation, pattern recognition receptor (PRR) ligands and endogenous immune activators can collaborate with IFNγ to induce the expression of co-inhibitory molecules, like programmed death-ligand 1 (PD-L1) (*Yamazaki et al., 2002*; *Hu and Ivashkiv, 2009*; *Krawczyk et al., 2010*; *Liu et al., 2017*; *McAdam, 1998*; *Nau et al., 2002*; *Schnare et al., 2001*), which binds T cell programmed death receptor 1 (PD1) to limit immune activation and mitigate T cell-mediated tissue damage (*Francisco et al., 2010*; *Abbas and Sharpe, 2005*; *Brown et al., 2003*; *Schildberg et al., 2016*).

IFNγ mediates these complex effects via binding to a heterodimeric surface receptor (*Bousoik and Montazeri Aliabadi, 2018*; *Garcia-Diaz et al., 2017*; *Ealick et al., 1991*; *Pestka et al., 2004*). The subunits of the complex, IFNGR1 and IFNGR2, assemble once IFNGR1 is bound by its ligand (*Blouin and Lamaze, 2013*; *Lasfar et al., 2014*). Complex assembly promotes the phosphorylation of Janus kinases 1 and 2 (JAK1 and JAK2) followed by activation of the signal transducer and activation of transcription 1 (STAT1) (*Meraz et al., 1996*). Phosphorylated STAT1 then dimerizes and translocates to the nucleus to activate the transcription of genes containing promoters with IFNγ-activated sequences (GAS), which includes other transcription factors such as interferon regulatory factor 1 (*Irf1*) that amplify the expression of a large regulon that includes T cell co-signaling molecules (*Schroder et al., 2004*; *Lehtonen et al., 1997*). The importance of this signaling pathway is evident in a variety of diseases including cancer (*Chen et al., 2012*; *Walser et al., 2007*; *Lyford-Pike et al., 2013*; *Garrido et al., 1997*; *Beatty and Paterson, 2001*), autoimmunity (*Pollard et al., 2013*; *Lees and Cross, 2007*), and infection (*Bustamante et al., 2014*). Individuals with inborn deficiencies in IFNγ signaling, including mutations to the receptor (*Newport et al., 1996*; *Jouanguy et al., 1996*), suffer from a defect in Th1 immunity that results in an immunodeficiency termed Mendelian susceptibility to mycobacterial disease (MSMD) (*Alcaïs et al., 2005*; *Bogunovic et al., 2012*; *Filipe-Santos et al., 2006*; *Kong et al., 2013*). Conversely, antagonists of IFNγ-inducible inhibitory molecules, such as PD-L1, are the basis for checkpoint inhibitor therapies that effectively promote T cell-mediated tumor immunity (*Schildberg et al., 2016*; *Garcia-Diaz et al., 2017*; *Sharpe, 2017*; *Castro et al., 2018*; *George et al., 2017*; *Gong et al., 2019*; *Ivashkiv, 2018*; *Mahoney et al., 2019*). While the obligate components of the IFNγ signaling pathway are well known, characterization of additional regulators of this response promises to identify both additional causes of immune dysfunction and new therapeutic targets.

Recent data suggest that cellular metabolism is an important modulator of the APC-T cell interaction. In particular, microbial stimulation of PRRs on the APC induces glycolytic metabolism and this shift in catabolic activity is essential for cellular activation, migration, and CD4+ and CD8+ T cell activation (*Krawczyk et al., 2010*; *Guak et al., 2018*; *Balic et al., 2020*; *Carneiro et al., 2018*; *Everts et al., 2014*; *Everts et al., 2012*; *Jha et al., 2015*; *Liu et al., 2017 Mills et al., 2016*; *Jung et al., 2018*; *Palmieri et al., 2020*; *Wang et al., 2018*; *Baardman et al., 2018*; *Cheng et al., 2014*; *Mills et al., 2018*; *Tannahill et al., 2013*). The metabolic state of the T cell is also influenced by the local environment and determines both effector function and long-term differentiation into memory cells (*Veldhoen et al., 2018*; *Buck et al., 2015*). IFNγ stimulation has been reported to induce glycolysis, suppress the target of rapamycin complex 1 (mTORC1), and modulate both cellular metabolism and translation in macrophages (*Wang et al., 2018*; *Su et al., 2015*). However, the effects of different metabolic states on IFNγ-stimulated APC function remain unclear.

To globally understand the cellular pathways that influence IFNγ-dependent APC function, we used a CRISPR-Cas9 knockout (KO) library (*Doench et al., 2016*) in macrophages to perform three parallel forward-genetic screens for regulators of three IFNγ-inducible co-signaling molecules: MHC class II (MHCII), CD40, and PD-L1. We identified positive and negative regulators that controlled each marker, underscoring the complex regulatory networks that influence the interactions between APCs and T cells. Pooled analysis of the screens uncovered shared regulators that contribute to the global

IFNγ response. Prominent among these general regulators was complex I of the respiratory chain. We report that the activity of the IFNγ receptor complex and subsequent transcriptional activation depends on mitochondrial function in both mouse and human myeloid cells. Experimental perturbation of respiration inhibits the capacity of both macrophages and DCs to stimulate T cells, identifying mitochondrial function as a central point where local signals are integrated to determine APC function.

## Results

### Forward genetic screen identifies regulators of IFNγ-inducible MHCII, CD40, and PD-L1 cell surface expression

To investigate the diverse regulatory pathways underlying the IFNγ response, we examined the expression of three functionally distinct cell surface markers that are induced by IFNγ. These studies used a J2 virus transformed bone marrow-derived macrophage (BMDM) line that expressed Cas9 (*Kiritsy et al., 2021*). Stimulation of these macrophages with IFNγ for 24 hours resulted in the upregulation of T cell stimulatory molecules, MHCII and CD40, and the inhibitory ligand PD-L1 (*Cd274*), on the cell surface (*Figure 1A*). To identify genes that regulate the expression of these markers, Cas9-expressing macrophages were transduced with a lentiviral genome-wide KO library containing four single-guide RNAs (sgRNAs) per protein-coding gene and 1000 non-targeting control (NTC) sgRNAs (*Doench et al., 2016*). The KO library was then stimulated with IFNγ, and fluorescently activated cell sorting (FACS) was used to select mutants with high or low cell surface expression of each individual marker (*Figure 1B*). For each of the three surface markers, positive and negative selections were performed in duplicate. The sgRNAs contained in the input library and each sorted population were amplified and sequenced (*Figure 1A and B*).

To estimate the strength of selection on individual mutant cells, we specifically assessed the relative abundance of cells harboring sgRNAs that target each of the surface markers that were the basis for cell sorting. When the abundances of sgRNAs specific for *H2-Ab1* (encoding the MHCII, H2-I-A beta chain), *Cd40*, or *Cd274* (PD-L1) were compared between high- and low-expressing cell populations, we found that each of these sgRNAs were significantly depleted from the cell populations expressing the targeted surface molecule, while each had no consistent effect on the expression of non-targeted genes (*Figure 1C*). While not all individual sgRNAs produced an identical effect, we found that targeting the genes that served as the basis of sorting altered the mean relative abundance 30–60-fold, demonstrating that all selections efficiently differentiated responsive from non-responsive cells.

We next tested for statistical enrichment of sgRNAs using MAGeCK-MLE (*Li et al., 2015*), which employs a generalized linear model to identify genes, and by extension regulatory mechanisms, controlling the expression of each surface marker. This analysis correctly identified the differential representation of sgRNAs targeting genes for the respective surface marker in the sorted populations in each screen, which were found in the top 20 ranked negative selection scores (ranks: *H2-Ab1* = 20, *Cd40* = 1, *Cd274* = 3; *Figure 1—source data 1*). Upon unsupervised clustering of selection coefficients determined by MAGeCK (β scores) for the most highly enriched genes in each screen (top 5%, positive or negative), both common and pathway-specific effects were apparent (*Figure 1D*, *Figure 1—source data 2*). A small number of genes assigned to cluster 1, including the IFNγ receptor components (*Ifngr1* and *Ifngr2*), were strongly selected in the non-responsive population of all three selections. However, many mutations appeared to preferentially affect the expression of individual surface markers, including a number of known pathway-specific functions. For example, genes previously shown to specifically control MHCII transcription, such as *Ciita*, *Rfx5*, *Rfxap*, *Rfxank*, and *Creb1* (*Steimle et al., 1994*; *Ferwerda et al., 2005*; *Chapoval et al., 2001*; *Steimle et al., 1995*), were found in cluster 4 along with several novel regulators that appear to be specifically required for this pathway. MHCII-specific factors are reported in an accompanying study (*Kiritsy et al., 2021*).

Genes specifically required for CD40 expression in cluster 3 included the heterodimeric receptor for TNF. *Tnfrsf1a* and *Tnfrsf1b* were the 6th and 50th lowest β scores in the CD40 screen, respectively. Previous studies suggested that TNF stimulation enhances IFNγ-mediated CD40 expression in hematopoietic progenitors (*Gu et al., 2012*), and we confirmed this observation in macrophages (*Figure 1E*). We observed a six-fold higher induction of CD40 in macrophages stimulated with a combination of IFNγ and TNF compared to IFNγ alone. This synergy was specific to CD40 induction as we did not observe any enhancement of IFNγ-induced MHCII expression by TNF addition. While

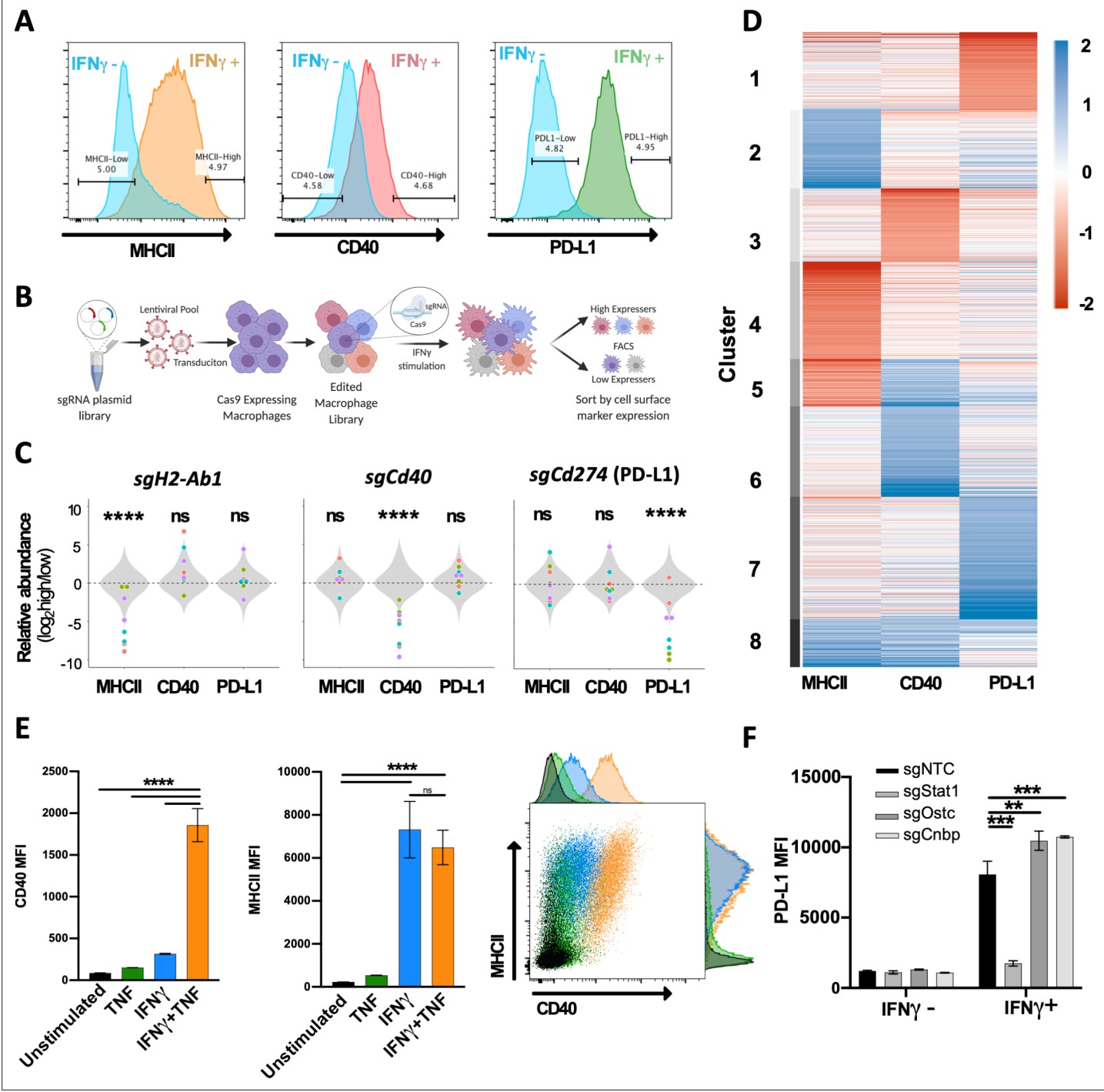

**Figure 1.** Forward genetic screen to identify regulators of the IFNγ response. (**A**) Representative histograms of the three selected cell surface markers targeted in macrophage CRISPR screens: MHCII, CD40, and PD-L1. Blue histograms indicate expression of each marker in unstimulated macrophages, and alternatively colored histograms show expression following 24 hr stimulation with recombinant murine IFNγ (10 ng/mL). Gates used for sorting 'high' and 'low' populations are shown. (**B**) Schematic of CRISPR screens. (**C**) Relative enrichment of select positive control (points) and all 1000 non-targeting control (NTC) sgRNAs (gray distribution) are plotted as a function of their log2 fold enrichment ('high' vs. 'low' bins). Data are from both replicate selections for each sgRNA (sgRNA denoted by shape). (**D**) Heatmap of β scores from CRISPR analysis, ordered according to k-means clustering (k = 8) of the 5% most enriched or depleted genes in each screen. (**E**) Macrophages were stimulated for 24 hr with TNF (25 ng/mL), IFNγ (10 ng/mL), or both TNF and IFNγ. Mean fluorescence intensity (MFI) of CD40 and MHCII was quantified by flow cytometry. Data are mean ± standard deviation for three biological replicates. Representative scatter plot from two independent experiments is provided. (**F**) Macrophages transduced with sgRNA targeting *Stat1, Ostc, Cnbp,* or a NTC were cultured with or without IFNγ for 24 hr, and cell surface expression of PD-L1 (MFI) was quantified by flow

*Figure 1 continued on next page*

*Figure 1 continued*

cytometry. For each genotype, data are the mean of cell lines with two independent sgRNAs ± standard deviation. Data are representative of three independent experiments. Statistical testing in panel (**C**) was performed with Tukey's multiple comparisons test. Within each screen, the sgRNA effects for each gene were compared to the distribution of NTC sgRNAs. Statistical testing in panels (**E**) and (**F**) was performed by one-way ANOVA with Holm–Sidak multiple comparisons correction. p-Values of 0.05, 0.01, 0.001, and 0.001 are indicated by *, **, *** and ****, respectively.

The online version of this article includes the following figure supplement(s) for figure 1:

**Source data 1.** Whole-genome profiling of macrophage MHCII, CD40, and PD-L1 expression.

**Source data 2.** k-means clustering of CRISPR-KO beta scores for regulators of the IFNγ response.

these results do not define the full TNF-related signaling pathway, they are consistent with the specific association between TNF receptor expression and CD40 induction.

Several recent studies have identified genes that control PD-L1 expression in cancer cell lines (*Garcia-Diaz et al., 2017*; *Gong et al., 2019*; *Mahoney et al., 2019*; *Kataoka et al., 2016*; *Burr et al., 2017*; *Coelho et al., 2017*; *Manguso et al., 2017*; *Mezzadra et al., 2017*; *Hassounah et al., 2019*), and we validated the PD-L1-associated clusters using these candidates. Our analysis found the previously described negative regulators, Irf2 (*Kriegsman et al., 2019*), Keap1, and Cul3 (*Papalexi et al., 2020*; *Wang et al., 2019*; *Wijdeven et al., 2018*), in the PD-L1-related cluster 7, along with novel putative negative regulators such as the oligosaccharyltransferase complex subunit *Ostc* and the transcriptional regulator, *Cnbp*. We generated KO macrophages for each of these novel candidates and confirmed that mutation of these genes enhances the IFNγ-dependent induction of PD-L1 surface levels (*Figure 1F*). Cumulatively, these data delineate the complex regulatory networks that shape the IFNγ response.

## Mitochondrial complex I is a positive regulator of the IFNγ response

To identify global regulators of the IFNγ response, we performed a combined analysis, reasoning that treating each independent selection as a replicate measurement would increase our power to identify novel pathways. We again used MAGeCK to calculate a selection coefficient (β) for each gene by maximum likelihood estimation (*Li et al., 2015*). By combining the 24 available measurements for each gene (three different markers, each selection in duplicate, and four sgRNAs per gene), we found that the resulting selection coefficient reflected the global importance of a gene for the IFNγ response (*Figure 2—source data 1*). The most important positive regulators corresponded to the proximal IFNγ signaling complex (*Figure 2A*). Similarly, we identified known negative regulators of IFNγ signaling, including the protein inhibitor of activated Stat1 (*Pias1*) (*Liu et al., 1998*), protein tyrosine phosphatase non-receptor type 2 (*Ptpn2*) (*Manguso et al., 2017*), mitogen activate protein kinase 1 (*Mapk1*), and suppressor of cytokine signaling 1 (*Socs1*) and 3 (*Socs3*).

We performed gene set enrichment analysis (GSEA) using a ranked list of positive regulators from the combined analysis (*Figure 2—source data 2*; *Subramanian et al., 2005*). Among the top enriched pathways was a gene set associated with type II interferon (e.g., IFNγ) signaling (normalized enrichment score = 2.45, q-value = 7.98e-5) validating the approach. GSEA identified a similarly robust enrichment for gene sets related to mitochondrial respiration and oxidative phosphorylation (*Figure 2B*). In particular, we found a significant enrichment of gene sets dedicated to the assembly and function of the NADH:ubiquinone oxidoreductose (hereafter, 'complex I') of the mitochondrial respiratory chain. Complex I couples electron transport with NADH oxidation and is one of four protein complexes that comprise the electron transport chain (ETC) that generates the electrochemical gradient for ATP biosynthesis. To confirm the GSEA results, we examined the combined dataset for individual genes that make up each complex of the ETC (*Figure 2C*). This analysis failed to demonstrate a clear role for sgRNAs targeting components of complexes II, III, or IV in the expression of the IFNγ-inducible surface markers tested. In contrast, the disruption of almost every subunit of complex I impaired the response to IFNγ, with the notable exception of *Ndufab1*. As this gene is essential for viability (*Stroud et al., 2016*), we assume that cells carrying *Ndufab1* sgRNAs retain functional target protein.

To investigate the contribution of specific complex I components to different IFNγ-stimulated phenotypes, we reviewed the surface marker-specific enrichment scores for genes that contribute to the complex assembly, the electron-accepting N-module, or the electron-donating Q module (*Stroud et al., 2016*; *Lazarou et al., 2007*; *Pagliarini et al., 2008*; *Baradaran et al., 2013*; *Zickermann et al.,*

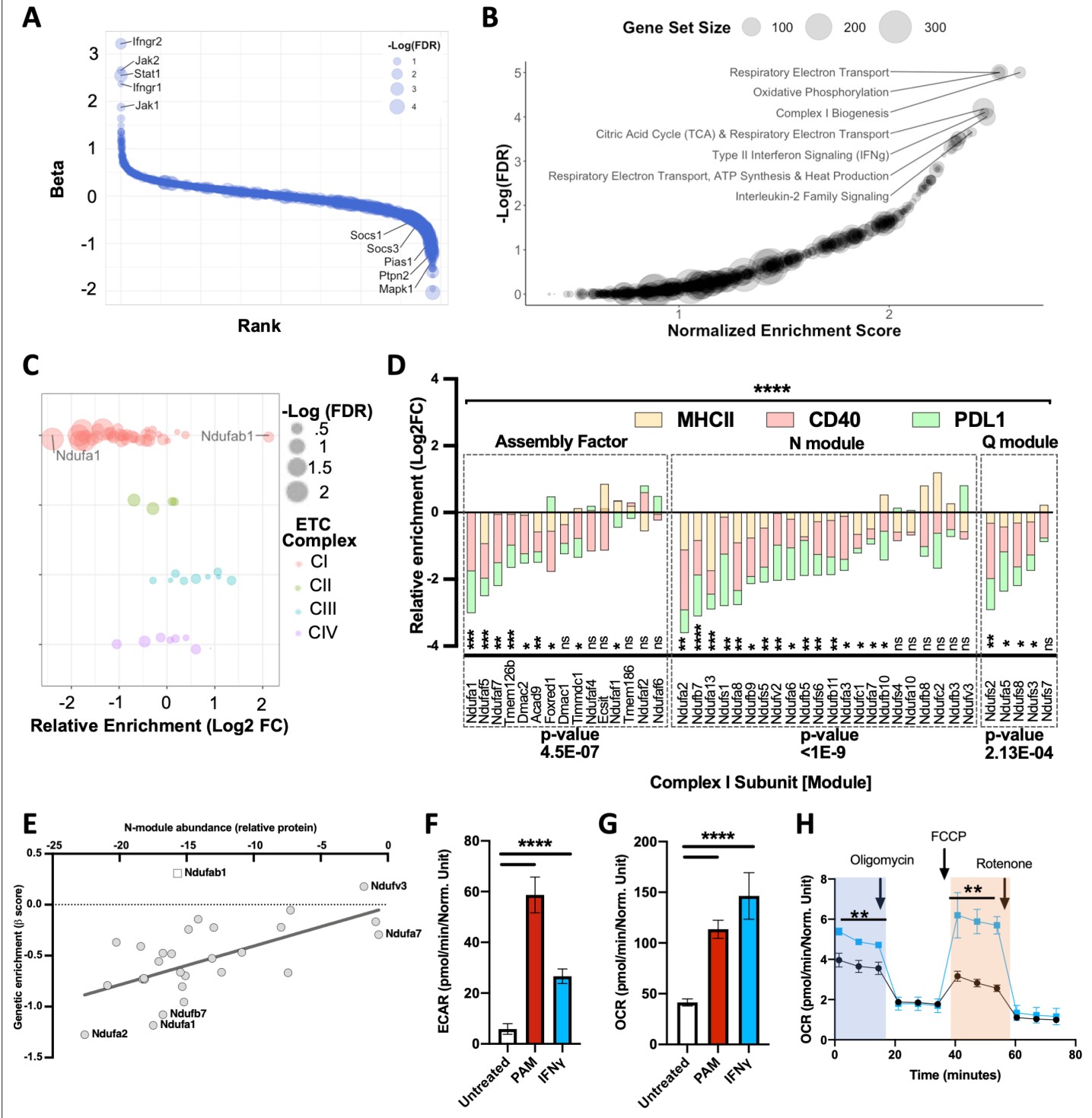

**Figure 2.** Global analysis of knockout (KO) libraries implicates mitochondrial complex I is a positive regulator of the IFNγ response. (**A**) Rank plot of the combined analysis for all genome-wide KO screens. Gene ranks (x-axis) were determined by maximal likelihood estimation (MLE). Known positive (left) and negative (right) regulators of IFNγ-mediated signaling are highlighted. The q-value (false discovery rate [FDR]) for each gene is indicated by dot size (-log$_{10}$ FDR). (**B**) Gene set enrichment analysis (GSEA) is based on the ranked list of positive regulators. Non-redundant pathways with a normalized enrichment score (NES) exceeding 2.0 and an FDR below 0.025 are labeled. (**C**) Relative enrichment (log2 fold change between 'high' and "low" bins) of genes that comprise the mitochondrial respirasome (GeneOntology 0005746) and were targeted in the CRISPR KO library. Respirasome components are grouped by electron transport chain (ETC) complex. FDR is based on MAGeCK-MLE. (**D**) Screen-specific enrichment score is plotted for complex I structural subunits and assembly factors. The statistical enrichment of a gene (e.g., *Ndufa1*) or module (e.g., N) was calculated using a binomial

*Figure 2 continued on next page*

*Figure 2 continued*

distribution function to calculate the probability that observed sgRNAs under examination would be depleted or enriched given the expected median probability. p-Values of 0.05, 0.01, 0.001, and 0.001 are indicated by *, **, ***, and ****, respectively. (**E**) Correlation between the relative effect of each complex I subunit on the structural integrity of the N-module (x-axis) with the relative requirement of each complex I subunit for the IFNγ response (y-axis; β score, as in panel **D**). The Pearson correlation coefficient (r) was calculated to be 0.6452 (95% confidence interval 0.3584–0.8207); p-value=0.0002. As *Ndufab1* (empty square) is an essential gene, its detection in the library indicates editing did not eliminate function; therefore, it was excluded from correlation analysis. (**F**) Following stimulation with IFNγ or PAM, extracellular acidification rate (ECAR)and (G) Oxygen consumption rate (OCR) values were measured by Seahorse in primary bone marrow-derived macrophages (BMDMs). Basal OCR and ECAR were determined 24 hr after stimulation with 10 ng/mL IFNγ or 200 ng/mL PAM. ****p<0.0001 by one-way ANOVA. (H) In a parallel experiment, the indicated chemical modulators were added to resting (Black) or IFNγ-activated (Blue) BMDMs at the indicated time points after initiating metabolic monitoring and the OCR response was monitored. Basal OCR (blue box) and maximal OCR (red box) are highlighted (right panel). **p<0.01 by two-tailed t-test.

The online version of this article includes the following figure supplement(s) for figure 2:

**Source data 1.** Whole-genome profiling of the IFNγ response in macrophages.

**Source data 2.** Gene set enrichment analysis (GSEA) to identify pathways that regulate the IFNγ response in macrophages.

*2015*; *Zhu et al., 2016*). Of the 42 individual assembly factors or structural subunits of complex I present in our mutant library, 29 were significantly enriched as positive regulators in the global analysis and were generally required for the induction of all IFNγ-inducible markers (*Figure 2D*). The enrichment for each functional module in non-responsive cells was statistically significant. However, not all individual complex I components were equally enriched, which could reflect either differential editing efficiency or distinct impacts on function. To investigate the latter hypothesis, we compared our genetic data with a previous proteomic study that quantified the effect of individual complex I subunits on the stability of the largest subcomplex, the N-module (*Stroud et al., 2016*). For a given subunit, we found a significant correlation between the magnitude of enrichment in our genetic screen and its effect on the structural stability of the module (*Figure 2E*), specifically implicating the activity of complex I in the IFNγ response.

These genetic data suggested a role for oxidative phosphorylation (OXPHOS) in the IFNγ response. To verify the mechanism of energy generation utilized by IFNγ-treated cells, we measured both their OXPHOS-dependent oxygen consumption rate (OCR) and glycolysis-dependent extracellular acidification rate (ECAR). IFNγ or toll-like receptor (TLR) 2 stimulation with Pam3CSK4 produced similar overall metabolic effects, increasing both OCR and ECAR in these cells (*Figure 2F,G*). Further analysis of mitochondrial function revealed that IFNγ increased both the basal and maximal OCR that was observed upon decoupling electron transport from ATP generation (*Figure 2H*) with carbonyl cyanide-4-trifluoromethoxy phenylhydrazone (FCCP).

To directly test the role of OXPHOS in the IFNγ response, we used CRISPR to generate individual macrophage lines that were deficient for complex I subunits. We first validated the expected metabolic effects of complex I disruption by comparing the intracellular ATP levels in macrophages carrying non-targeting control sgRNA (sgNTC) with sg*Ndufa1* and sg*Ndufa2* lines. When cultured in media containing the glycolytic substrate, glucose, all cell lines produced equivalent amounts of ATP (*Figure 3A*). However, when pyruvate was provided as the sole carbon source and ATP generation depended entirely upon flux through ETC and OXPHOS, both sg*Ndufa1* and sg*Ndufa2* macrophages contained decreased ATP levels compared to sgNTC cells (*Figure 3B*). To confirm the glycolytic dependency of complex I mutant macrophages, we grew cells in complete media with glucose and treated with the ATP synthase (complex V) inhibitor, oligomycin, which blocks ATP generation by OXPHOS. While oligomycin reduced ATP levels in sgNTC macrophages, this treatment had no effect in sg*Ndufa1* and sg*Ndufa2* cells (*Figure 3—figure supplement 1A*), confirming that these complex I-deficient cells rely on glycolysis for energy generation. IFNγ treatment slightly reduced ATP levels in glucose-containing media but did not differentially affect cell lines (*Figure 3A*). Throughout these experiments, we found that the sg*Ndufa1* mutant showed a greater OXPHOS deficiency than the sg*Ndufa2* line.

We next compared the response to IFNγ in macrophages lacking *Ndufa1* and *Ndufa2* with those carrying CRISPR-edited alleles of *Ifngr1* or the negative regulator of signaling, *Ptpn2*. As CD40 was found to rely on more complex inputs for expression, which include TNF (*Figure 1E*), we relied on MHCII and PD-L1 as markers of the IFNγ response for subsequent studies. As expected, and consistent with the genetic screen, we found that the loss of *Ifngr1* or *Ptpn2* either abrogated or

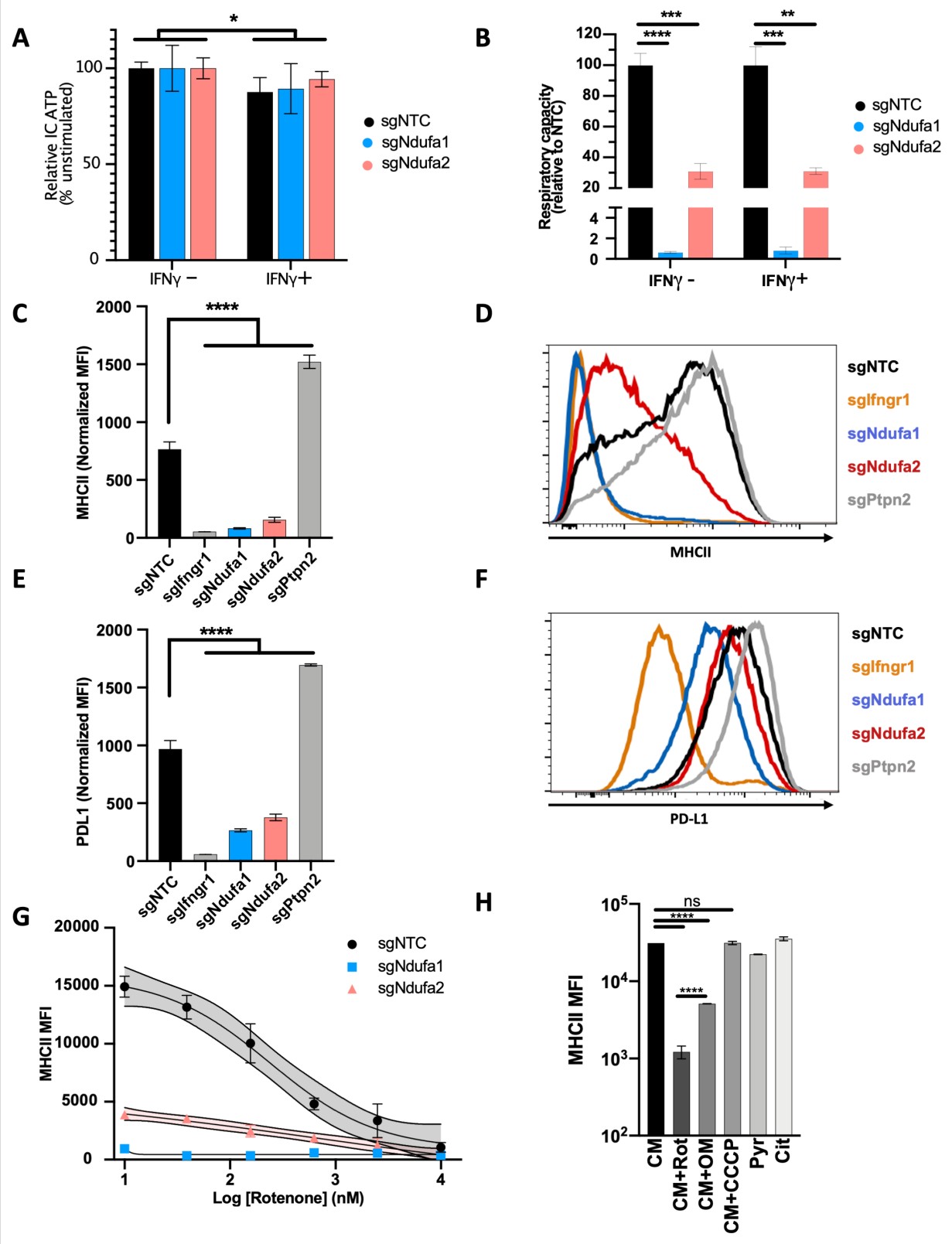

**Figure 3.** Complex I is necessary for IFNγ-induced MHCII and PD-L1 expression. Metabolic phenotypes in macrophage mutants were confirmed by measuring intracellular (IC) ATP abundance following culture in media containing only (**A**) glucose or (**B**) pyruvate. Values are normalized to the average respiratory capacity of non-targeting control macrophages (NTC) and are the mean ± standard deviation for four biological replicates. Statistical testing within each condition (with or without IFNγ for 24 hr) was performed by one-way ANOVA with Dunnett's multiple comparisons correction.

*Figure 3 continued on next page*

*Figure 3 continued*

(**C–F**) NTC, positive control (sg*Ifngr1* and sg*Ptpn2*), and complex I mutant (sg*Ndufa1* and sg*Ndufa2*) macrophages were stimulated for 24 hr with recombinant murine IFNγ. Plotted values in (**C**) and (**E**) are the geometric mean fluorescence intensity (MFI) for a given mutant normalized to an internal control present in each well; for each gene, the data are the mean for two independent sgRNAs ± standard deviation. Representative histograms are provided in (**D**) and (**F**). Data are representative of >5 independent experiments. (**G**) MHCII MFI of macrophages stimulated with IFNγ and treated with rotenone at the indicated concentrations for 24 hr. Mean ± standard deviation for two biological replicates are shown. Data are representative of four independent experiments. (**H**) Left: MHCII MFI on macrophages cultured in complete media (CM) and stimulated with IFNγ and the indicated inhibitors for 24 hr. Right: MHCII MFI on macrophages cultured in CM or media containing only pyruvate (Pyr) or citrate (Cit) stimulated with IFNγ for 24 hr. Mean ± standard deviation for two or three biological replicates is indicated. Data are representative of four independent experiments. Statistical testing was performed by one-way ANOVA with Tukey's correction for multiple hypothesis testing. p-Values of 0.05, 0.01, 0.001, and 0.001 are indicated by *, **, ***, and ****, respectively.

The online version of this article includes the following figure supplement(s) for figure 3:

**Figure supplement 1.** The complex I dependency of IFNγ signaling is independent of reactive oxygen and nitrogen radicals, and Hif-1α.

enhanced the response to IFNγ, respectively. Also consistent with predictions, mutation of complex I genes significantly reduced the IFNγ-dependent induction of MHCII and PD-L1 compared to sgNTC (*Figure 3C–F*). The *Ndufa1* mutation that abolishes OXPHOS reduced MHCII induction to the same level as *Ifngr1*-deficient cells. To confirm these results using an orthologous method, we treated cells with the complex I inhibitor, rotenone (*Barrientos and Moraes, 1999*). This treatment caused a dose-dependent inhibition of the IFNγ-induced MHCII expression in sgNTC macrophages (*Figure 3G*) and had a similar inhibitory effect on the residual IFNγ response in *Ndufa2*-deficient cells. Together, these results confirm that complex I is required for the induction of immunomodulatory surface molecules in response to IFNγ.

To determine which aspect of mitochondrial respiration contributes to the IFNγ response, we inhibited different components of the ETC. Rotenone, oligomycin, and carbonyl cyanide m-chloro-phenyl hydrazone (CCCP) were used at concentrations that abolished the OXPHOS-dependent ATP generation that is necessary in pyruvate media (*Figure 3—figure supplement 1B*). The complex V inhibitor, oligomycin, inhibited the IFNγ-induced MHCII expression, albeit to a lesser extent than direct complex I inhibition with rotenone (*Figure 3H*). This partial effect could reflect an inability to dissipate the proton motive force (PMF), which inhibits electron flux throughout the ETC, including through complex I (*Brand and Nicholls, 2011*). CCCP disrupts mitochondrial membrane potential and OXPHOS while preserving electron flux. CCCP had no effect on the IFNγ response, indicating that ATP generation is dispensable for IFNγ responsiveness and highlighting a specific role for complex I activity. To directly test the contributions of complex III and IV to the IFNγ response, we inhibited each complex using antimycin A and sodium azide, respectively. While treatment with these inhibitors lowered ATP concentration similarly to rotenone in glucose-containing media (*Figure 3—figure supplement 1C*), the effect of complex III or IV inhibition on IFNγ-mediated MHCII expression was less than rotenone treatment and was similar to the partial inhibition observed with oligomycin (*Figure 3—figure supplement 1D*). These observations are consistent with the initial screen, where complex I inhibition produced the most pronounced effect on the IFNγ response.

We then altered the media composition to test the sufficiency of mitochondrial respiration to drive IFNγ responses independently from aerobic glycolysis. IFNγ was found to stimulate MHCII expression to a similar degree in macrophages cultured in complete media with glucose as in media containing only pyruvate or citrate, which must be catabolized via mitochondrial respiration (*Figure 3H*). Taken together these results suggest that cellular respiration is both necessary and sufficient for maximal expression of the IFNγ-inducible surface markers MHCII and PD-L1.

## Mitochondrial function is specifically required for IFNγ-dependent responses

The mitochondrial dependency of the IFNγ response contrasted with the known glycolytic dependency of TLR signaling, suggesting that TLR responses would remain intact when complex I was inhibited. Indeed, not only were TLR responses intact in sg*Ndufa1* and sg*Ndufa2* mutant macrophages, these cells secreted larger amounts of TNF or interleukin 6 (IL-6) than sgNTC cells in response to the TLR2 ligand, Pam3CSK4. (*Figure 4A*). Thus, the glycolytic dependency of these cells enhanced the TLR2 response, indicating opposing metabolic dependencies for IFNγ and TLR signaling.

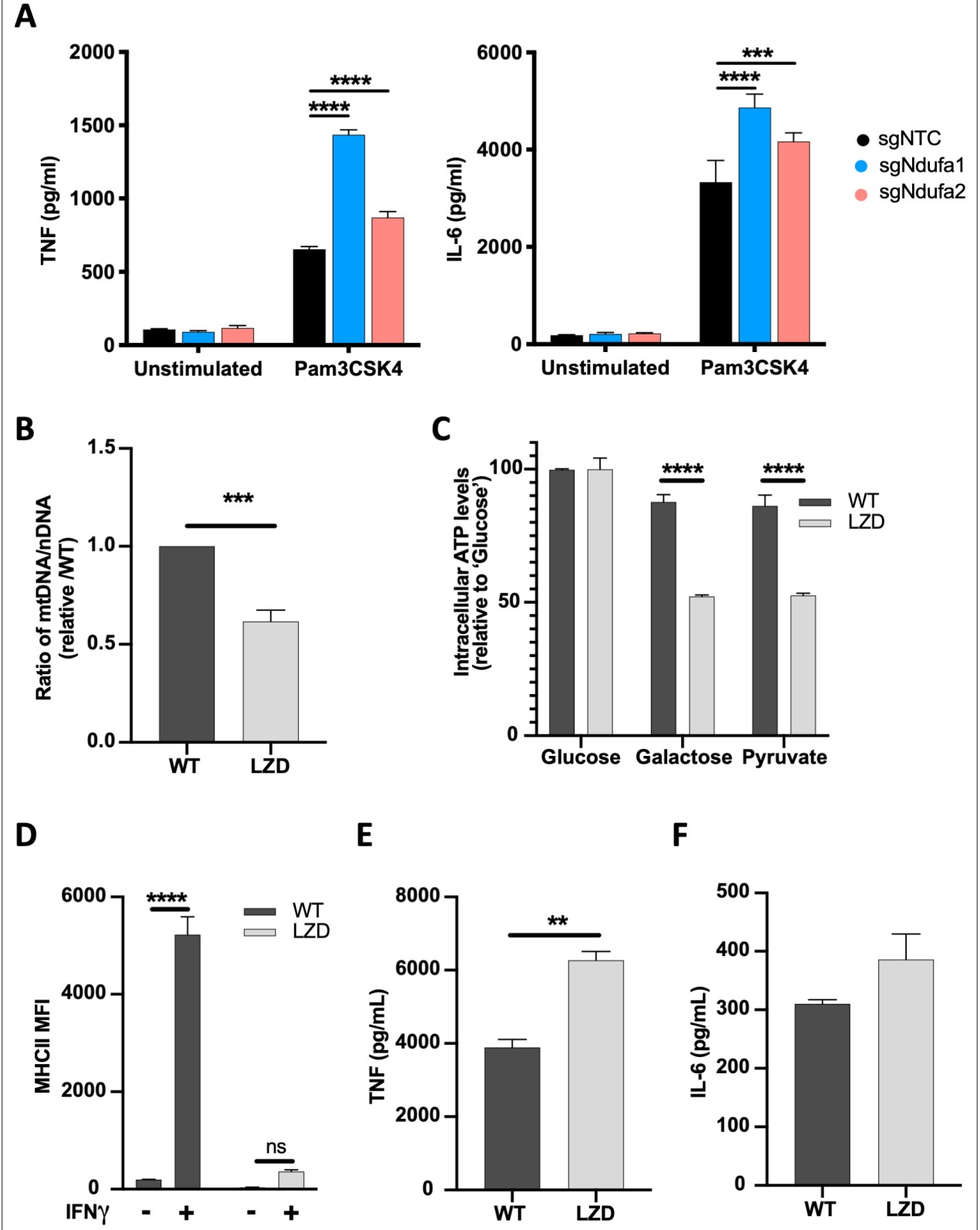

**Figure 4.** Diminished mitochondrial function specifically limits IFNγ-dependent responses. (**A**) TNF and IL-6 production by non-targeting control (NTC) or complex I mutant macrophages stimulated with Pam3CSK4 for 24 hr was determined by ELISA. Statistical testing between mutant and NTC macrophages from triplicate samples was performed by ANOVA with Dunnett's correction for multiple comparisons. Data are representative of two independent experiments. (**B**) qPCR determination of relative mitochondrial genomes present per nuclear genome in macrophages cultured in vehicle

*Figure 4 continued on next page*

*Figure 4 continued*

(WT) or 50 µg/mL linezolid (LZD). $C_t$ values were normalized to reference nuclear gene hexokinase 2 (*Hk2*) and plotted as abundance relative to WT. Data were analyzed by two-way unpaired t-test. (**C**) ATP abundance in control or LZD-conditioned macrophages cultured in 10 mM glucose, galactose, or pyruvate. ATP values normalized to mean of 10 mM glucose and plotted as percent. Mean ± standard deviation for two biological replicates of each condition. Differences were tested by two-way ANOVA using the Sidak method to correct for multiple hypothesis testing. (**D**) Mean fluorescence intensity (MFI) of MHCII was determined by flow cytometry on control or LZD-conditioned macrophages following 24 hr stimulation with IFNγ. Mean ± standard deviation for two biological replicates of each condition and representative of two independent experiments. Differences were tested by two-way ANOVA using Tukey's method to correct for multiple hypothesis testing. (**E, F**) Secretion of TNF and IL-6 in WT and LZD-conditioned macrophages following Pam3CSK4 stimulation for 6 hr was quantified by ELISA. Mean ± standard deviation for three biological replicates of each condition and two independent experiments. Data were analyzed by two-way unpaired t-test. p-Values of 0.05, 0.01, 0.001, and 0.001 are indicated by *, **, ***, and ****, respectively.

Whether the effect of complex I on macrophage responsiveness was the result of reduced mitochondrial respiratory function or secondary to cellular stress responses, such as radical generation, remained unclear. To more directly relate mitochondrial function to these signaling pathways, we created cell lines with reduced mitochondrial mass. Macrophages were continuously cultured in linezolid (LZD), an oxazolidinone antibiotic that inhibits the mitochondrial ribosome (*De Vriese et al., 2006*; *Soriano et al., 2005*; *Wilson et al., 2008*). This treatment produced a cell line with ~50% fewer mitochondrial genomes per nuclear genome and a corresponding decrease in OXPHOS capacity compared to control cells grown in the absence of LZD (*Figure 4B and C*). Cells were cultured without LZD for 16 hr and then stimulated with either IFNγ or Pam3CSK4. Consistent with our complex I inhibition studies, we found that this reduction in mitochondrial mass nearly abrogated the IFNγ-dependent induction of MHCII (*Figure 4D*), while the TLR2-dependent secretion of TNF and IL-6 was preserved or enhanced (*Figure 4E and F*). Thus, mitochondrial activity, itself, is necessary for a robust IFNγ response.

To further address potential secondary effects of mitochondrial inhibition on the IFNγ response, we investigated the role of known oxygen or nitrogen radical-dependent regulators (*Figure 3— figure supplement 1E–I*). Inhibition of ROS generation by replacing glucose with galactose (*Wang et al., 2018*; *Brand and Nicholls, 2011*; *Bustamante et al., 1978*) had no effect on IFNγ-induced MHCII induction. Similarly, neutralization of cytosolic or mitochondrial radicals with N-acetyl cysteine (NAC) or MitoTEMPO, respectively, had no effect on MHCII induction either alone or in combination with ETC inhibition. The role of the cytosolic redox sensor, HIF-1α (*Cramer et al., 2003*; *Semenza, 2012*), was addressed by chemically stabilizing this factor with dimethyloxalylglycine (DMOG). A potential role for nitric oxide production was addressed with the specific NOS2 inhibitor 1400W (*Everts et al., 2012*; *Wang et al., 2018*; *Braverman and Stanley, 2017*). Neither of these treatments affected IFNγ-induced MHCII cell surface expression in the presence or absence of simultaneous Pam3CSK4, further supporting a direct relationship between mitochondrial respiratory capacity and the IFNγ response.

## Complex I is specifically required for IFNγ signaling in human cells

To understand the function of complex I during IFNγ stimulation in human cells, we used monocyte-derived macrophages (MDMs) from peripheral blood of healthy donors differentiated in the presence of GM-CSF. As in our mouse studies, we assessed the response of these cells to IFNγ or Pam3CSK4 by quantifying the abundance of IFNγ-inducible surface markers or cytokines that were optimized for human cells. Since HLA-DR is not strongly induced by IFNγ, we included ICAM1 in addition to CD40 and PD-L1 as surface markers. As seen in the murine model, rotenone inhibited the IFNγ-mediated induction of all three markers (*Figure 5A*). Similar results were obtained using CD14+ monocytes and M-CSF-derived macrophages (*Figure 5—figure supplement 1A*). TLR2 responses were assessed by the production of TNF and IL-1β. Upon Pam3CSK4 stimulation of GM-CSF-derived MDM, rotenone significantly enhanced the secretion of IL-1β and TNF (*Figure 5B and C*). While simultaneous treatment with both IFNγ and Pam3CSK4 produced the previously described inhibition of IL-1β (*Mishra et al., 2013*), rotenone still did not decrease the production of these TLR2-dependent cytokines. Thus, as we observed in mouse cells, complex I is specifically required for IFNγ signaling in human macrophages.

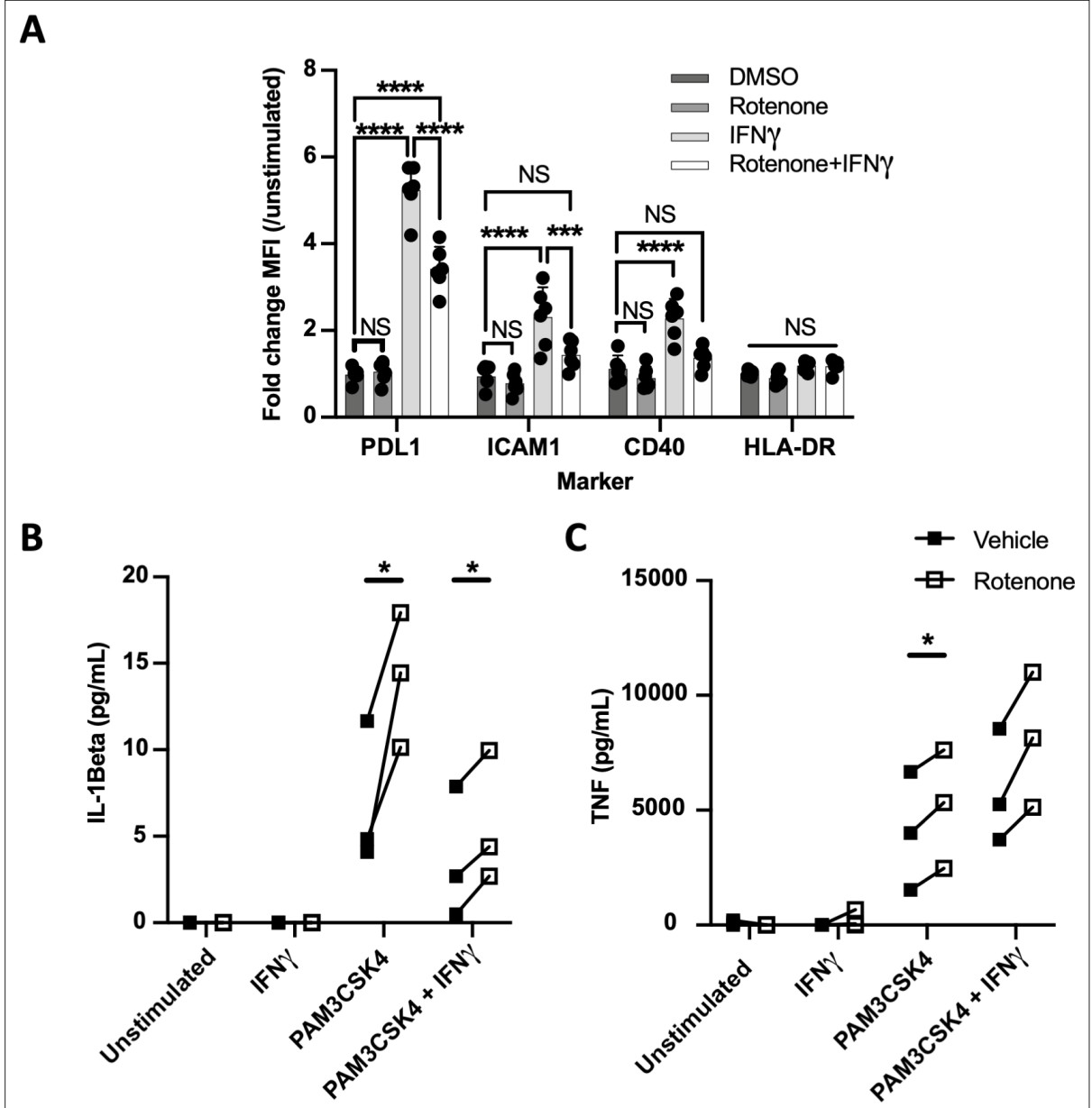

**Figure 5.** Complex I is specifically required for IFNγ signaling in human cells. (**A**) CD14+ monocytes from healthy human donors were differentiated into macrophages with GM-CSF. Mean fluorescence intensity (MFI) of cell surface markers PD-L1, ICAM1, CD40, and HLA-DR was determined by flow cytometry following stimulation with IFNγ and/or inhibition of complex I with rotenone (10 μM) for 24 hr. Data are representative of two independent experiments, and values are normalized to donor-specific unstimulated/vehicle control. Mean ± standard deviation for six biological replicates of each condition. Differences were tested by two-way ANOVA using the Sidak–Holm method to correct for multiple hypothesis testing. (**B, C**) Quantification of IL-1β and TNF production from primary human macrophages, measured by ELISA from cell supernatants following stimulation. Lines connect values for individual donors treated with vehicle (DMSO, black squares) or rotenone (empty squares). Differences were tested by repeated measures two-way ANOVA using the Sidak–Holm method to correct for multiple hypothesis testing. p-Values of 0.05, 0.01, 0.001, and 0.001 are indicated by *, **, ***, and ****, respectively.

The online version of this article includes the following figure supplement(s) for figure 5:

**Figure supplement 1.** Complex I is specifically required for IFNγ signaling in diverse human myeloid cells.

## Complex I inhibition reduces IFNγ receptor activity

To understand how complex I activity shapes the IFNγ response, we determined whether its effect was transcriptional or post-transcriptional by simultaneously monitoring mRNA and protein abundance over time. Surface expression of PD-L1 was compared with Cd274 mRNA abundance, while

the surface expression of MHCII was compared with the mRNA abundance of *Ciita*, the activator of MHCII expression that is initially induced by IFNγ (*Figure 6A and B*). In both cases, mRNA induction preceded surface expression of the respective protein. More importantly, both mRNA and protein expression of each marker were diminished to a similar degree in sg*Ndufa1* and sg*Ndufa2* compared to sgNTC cells. Thus, a deficit in transcriptional induction could account for the subsequent decrease in surface expression observed in complex I deficient cells.

IFNγ rapidly induces the transcription of a large number of STAT1 target genes, including *Irf1*, which amplifies the response. The relative impact of complex I inhibition on the immediate transcriptional response versus the subsequent IRF1-dependent amplification was initially assessed by altering the timing of complex I inhibition. As the addition of rotenone was delayed relative to IFNγ stimulation, the ultimate effect on MHCII expression was diminished (*Figure 6C*). If rotenone was added more than 4 hr after IFNγ, negligible inhibition was observed by 24 hr, indicating that early events were preferentially impacted by rotenone. To more formally test the role of IRF1, this study was performed in macrophages harboring a CRISPR-edited *Irf1* gene. While the level of MHCII induction was reduced in the absence of IRF1, the relative effect of rotenone addition over time was nearly identical in sg*Irf1* and sgNTC cells. Thus, mitochondrial function appeared to preferentially impact the initial transcriptional response to IFNγ upstream of IRF1.

Ligand-induced assembly of the IFNGR1-IFNGR2 receptor complex results in the phosphorylation and transactivation of janus kinases 1 and 2 (JAK1 and JAK2). Autophosphorylation of JAK2 at tyrosine residues 1007/1008 positively regulates this cascade and serves as a marker of JAK2 activation. These activating events at the cytoplasmic domains of the IFNγ receptor complex facilitate STAT1 docking and phosphorylation at tyrosine-701 (Tyr701), a prerequisite for the IFNγ response. Additional STAT1 phosphorylation at serine-727 (Ser727) can amplify signaling. To determine if complex I is required for these early signal transduction events, we examined the activation kinetics by immunoblot (*Figure 6D*). The total abundances of IFNGR1, STAT1, and JAK2, were constant in sgNTC and sg*Ndufa1* cells in the presence and absence of IFNγ stimulation. While we detected robust phosphorylation of JAK2 Y1007/8, STAT1-Y701, and STAT1-S727 over time following IFNγ treatment in sgNTC cells, phosphorylation at all three sites was both delayed and reduced across the time course in sg*Ndufa1* cells. We conclude that the loss of complex I function inhibits receptor proximal signal transduction events.

## Mitochondrial respiration in APCs is required for IFNγ-dependent T cell activation

As respiration affected both stimulatory and inhibitory APC functions, we sought to understand the ultimate effect of mitochondrial function on T cell activation. To this end, we generated myeloid progenitor cell lines from Cas9-expressing transgenic mice that can be used for genome-editing and differentiated into either macrophages or DCs using M-CSF or FLT3L, respectively (*Wang et al., 2006*; *Redecke et al., 2013*). Macrophages differentiated from these myeloid progenitors demonstrated robust induction of all three markers that were the basis for the IFNγ stimulation screens (*Figure 7—figure supplement 1A–C*). Further, both the IFNγ-mediated upregulation of these markers and the inhibitory effect of rotenone or oligomycin on their induction were indistinguishable from wild type primary BMDMs (*Figure 7—figure supplement 1D–F*). In both macrophages and in DCs, the induction of MHCII by IFNγ was inhibited by rotenone and oligomycin (*Figure 7A*). Unlike macrophages, murine DCs basally express MHCII and these inhibitors only repressed the further induction by IFNγ (*Figure 7A and B*).

Both macrophages and DCs were used to determine if the inhibition of complex I in APCs reduces T cell activation. Both types of APCs were stimulated with IFNγ overnight with or without rotenone before washing cells to remove rotenone and ensure T cell metabolism was unperturbed. APCs were then pulsed with a peptide derived from the *Mycobacterium tuberculosis* protein ESAT-6 and co-cultured with ESAT-6-specific CD4+ T cells from a TCR transgenic mouse (*Gallegos et al., 2008*). T cell activation was assayed by intracellular cytokine staining for IFNγ. In macrophages, T cell stimulation relied on pretreatment of the APC with IFNγ as a macrophage line lacking the *Ifngr1* gene was unable to support T cell activation. Similarly, inhibition of complex I in macrophages completely abolished antigen-specific T cell stimulation (*Figure 7C*). DCs did not absolutely require IFNγ pretreatment to stimulate T cells, likely due to the basal expression of MHCII by these cells.

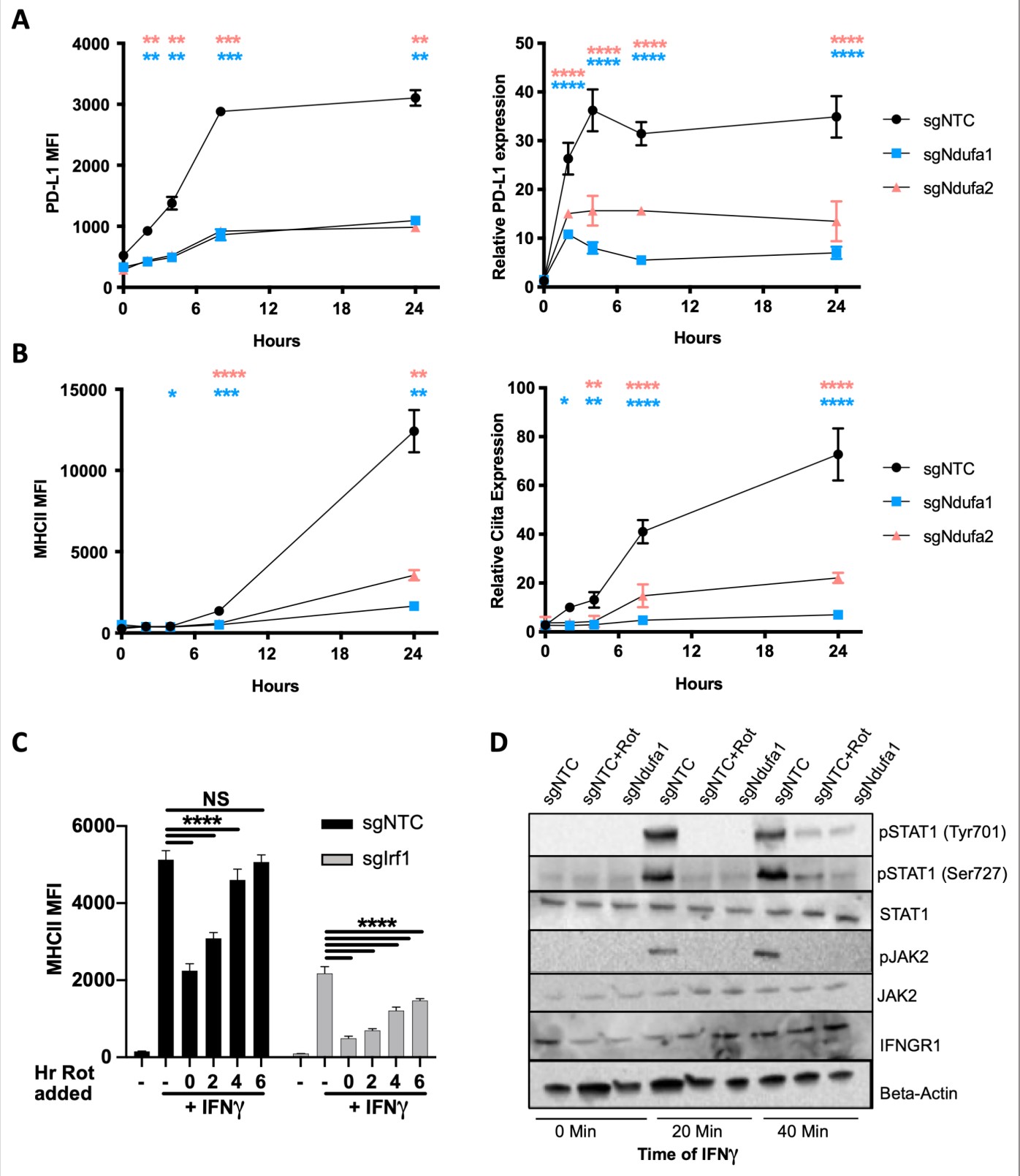

**Figure 6.** Complex I inhibition reduces IFNγ receptor activity. (**A**) PD-L1 transcript was quantified by qRT-PCR using ΔΔCt relative to *β-actin* in macrophages of the indicated genotype after stimulation with 10 ng/mL IFNγ. PD-L1 mean fluorescence intensity (MFI) was determined at the same time points by flow cytometry. (**B**) *Ciita* transcript was quantified by qRT-PCR using ΔΔCt relative to *β-actin Gapdh* in macrophages of the indicated genotype after stimulation with 10 ng/mL IFNγ. MHCII MFI was determined at the same time points by flow cytometry. Data shown are from biological

*Figure 6 continued on next page*

*Figure 6 continued*

triplicate samples with technical replicates for RT-PCR experiments and are representative of two independent experiments. (**C**) sgNTC (left) or sg*Irf1* (right) macrophages were cultured for 24 hr with or without IFNγ stimulation. At 2 hr intervals post-IFNγ stimulation, rotenone was added. After 24 hr of stimulation, cells were harvested and surface expression of MHCII (MFI) was quantified by flow cytometry. Data are mean ± standard deviation for three biological replicates and are representative of two independent experiments. Statistical testing was performed by one-way ANOVA with Tukey's correction for multiple hypothesis testing. (**D**) Control (non-targeting control [NTC]) or sg*Ndufa1* macrophages were stimulated with IFNγ for the indicated times while NTC macrophages were pretreated with 10 μM rotenone for 2 hr prior to IFNγ stimulation. Cell lysates analyzed by immunoblot for STAT1 abundance and phosphorylation (Y701 and S727), JAK2 abundance and phosphorylation (Y1007/8), and IFNGR1. β-Actin was used as a loading control. Data are representative of three independent experiments. Results shown are from a single experiment analyzed on three parallel blots. p-Values of 0.05, 0.01, 0.001, and 0.001 are indicated by *, **, ***, and ****, respectively.

The online version of this article includes the following figure supplement(s) for figure 6:

**Source data 1.** Raw blots.

**Source data 2.** Labeled raw blots.

Regardless, rotenone treatment of DCs abrogated the IFNγ-dependent increase in T cell stimulation (**Figure 7C**).

To confirm the effects of complex I inhibition on T cell activation using a genetic approach and confirm that complex I inhibition acted in a cell-autonomous mechanism, we generated *Ndufa1* KO myeloid progenitors (Hox-sg*Ndufa1*). Following differentiation into macrophages, Hox-sg*Ndufa1* demonstrated glycolytic dependence and an inability to generate ATP by OXPHOS compared to control Hox-sgNTC macrophages (**Figure 7—figure supplement 1G**). Having confirmed the expected metabolic effects of Ndufa1 loss, Hox-sg*Ndufa1* and Hox-sgNTC macrophages were mixed at various ratios. Mixed cultures were then stimulated with IFNγ, peptide pulsed, and co-cultured with antigen-specific CD4+ T cells. In agreement with our chemical inhibition studies, we found strong correlation between complex I activity in the APC population and T cell stimulatory activity (**Figure 7D and E**). Together, these data confirm that the IFNγ-dependent augmentation of T cell stimulatory activity depends on complex I function in both macrophages and DCs.

## Discussion

IFNγ-mediated control of APC function is central to shaping a protective immune response, and the canonical IFNγ signal transduction pathway has been elucidated in exquisite detail (**Bhat et al., 2018**). Our study demonstrates that unbiased genetic analyses can reveal a multitude of unexpected cellular regulators, even for a well-characterized process such as IFNγ signaling. By independently assessing genetic determinants of stimulatory and inhibitory molecule expression, we discovered mechanisms of regulation that preferentially affect the induction of different cell surface proteins. These results begin to explain how a single cytokine can induce functionally distinct downstream responses in different contexts. These data also suggest new strategies to modulate individual co-receptors to either stimulate or inhibit T cell activation. Another strength of our parallel screen approach was the increased power to identify shared mechanisms that control IFNγ-mediated regulation across all screens. Our pooled analysis identified mitochondrial respiration, and in particular complex I, as essential for IFNγ responses in APCs. We determined that complex I is required for the IFNγ-mediated induction of important co-signaling molecules and is necessary for antigen presentation and T cell activation. These findings uncover a new dependency between cellular metabolism and the immune response.

Both our work and others report that IFNγ stimulation alters macrophage metabolism. We found that IFNγ stimulation increases oxygen consumption and glycolytic activity, and previous studies found that this treatment mediates a gradual shift to aerobic glycolysis that might be related to a reduction in mTORC1 activity (**Wang et al., 2018**; **Su et al., 2015**). While the overall effect of IFNγ on cellular metabolism appears to be complex and change over time, our genetic data unequivocally reveal that mitochondrial respiration is required for IFNγ signaling. Data from the CRISPR screens suggested a preferential role for complex I, relative to complexes II, III, or IV. Further chemical inhibitor studies showed that blocking complex III, IV, or V (ATP synthase) does reduce IFNγ responses, but not to the same magnitude as complex I inhibition. These data could suggest that complex I is still able to support IFNγ signaling in the absence of these downstream complexes, perhaps by transferring

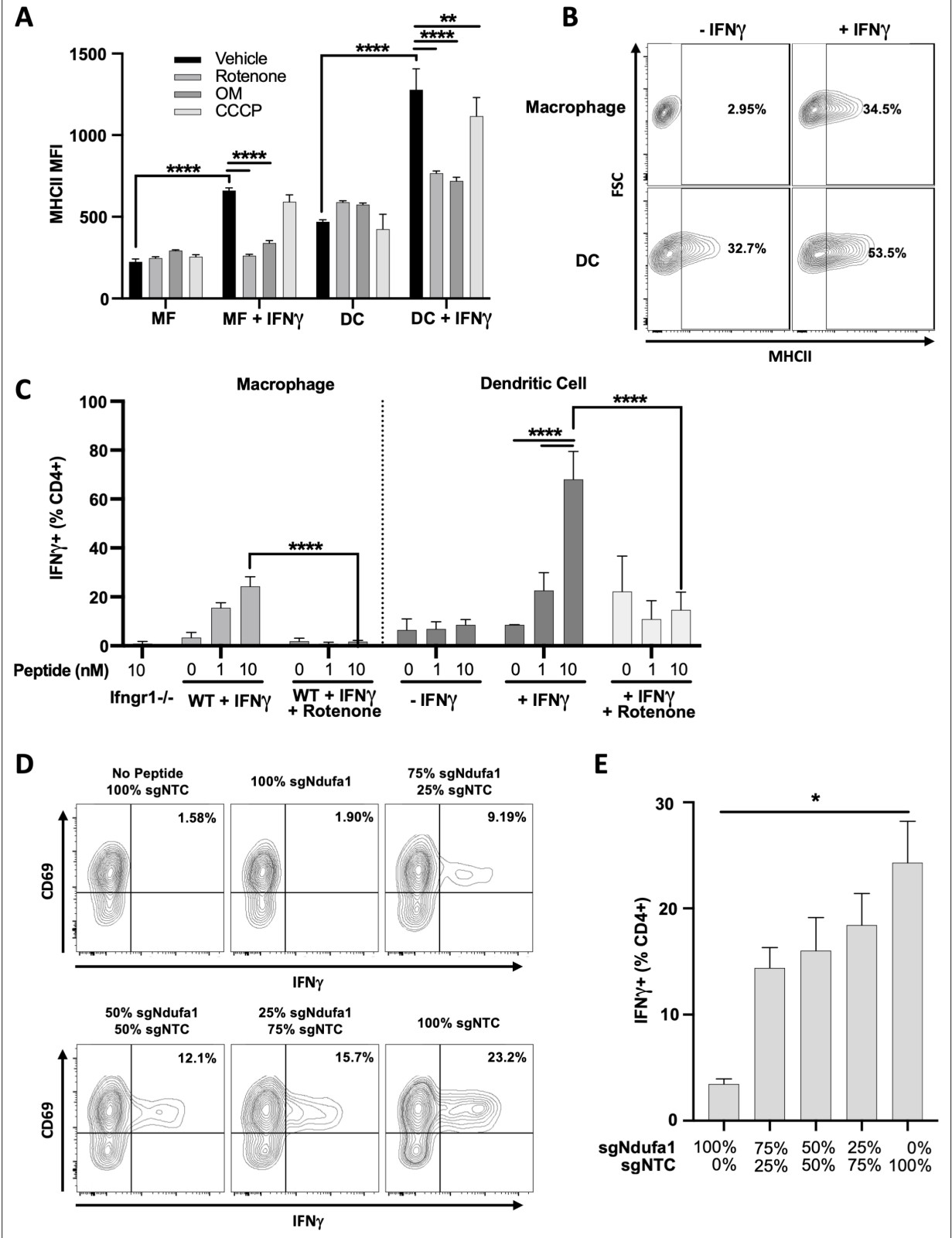

**Figure 7.** Mitochondrial respiration in antigen-presenting cells (APCs) is required for IFNγ-dependent T cell activation. (**A**) Cell surface expression of MHCII (mean fluorescence intensity [MFI]) in macrophages (MF) or dendritic cells (DCs) derived from conditionally immortalized progenitor lines. IFNγ was added for 24 hr where indicated. Cells were treated with vehicle (DMSO), rotenone (10 μM), oligomycin (OM, 2.5 μM), or carbonyl cyanide m-chlorophenyl hydrazone (CCCP) concurrent with IFNγ. Data are three biological replicates and are representative of at least two independent

*Figure 7 continued*

experiments. (**B**) Contour plot of macrophage (top row) or DC (bottom row) MHCII expression in the absence of (left column) or following (right column) stimulation with IFNγ for 24 hr. Representative samples were selected from (**A**). The percent MHCII positive are indicated for each of the conditions. (**C**) CD4+ T cell activation as measured by the percent of live cells positive for IFNγ by intracellular cytokine staining. Prior to co-culture with T cells, APCs were stimulated with the indicated combinations of IFNγ (10 ng/mL), and/or rotenone (10 µM) for 24 hr. After washing and pulsing with ESAT-61–15 at the indicated concentrations (nm), T cells were added to APCs at an effector to target (E:T) ratio of 1:1 and co-cultured for a total of 5 hr. Data are representative of two independent experiments. Data are mean ± standard deviation for three biological replicates. Statistical testing was performed by one-way ANOVA with Tukey's correction for multiple hypothesis testing. (**D, E**) sg*Ndufa1* or non-targeting control (NTC) macrophages were differentiated from immortalized progenitors and mixed at the ratios indicated (labeled as percent of knockout [KO] cells). Mixed cultures were stimulated with IFNγ for 24 hr, peptide loaded, and co-cultured with CD4+ T cells (E:T 1:1). Production of IFNγ was measured by ICS and quantified as the percent of cells positive for staining by flow cytometry. Representative contour plots (**D**) and quantification (**E**) of the experiment are shown. Data shown are for biological triplicate samples and are representative of two independent experiments. p-Values of 0.05, 0.01, 0.001, and 0.001 are indicated by *, **, ***, and ****, respectively.

The online version of this article includes the following figure supplement(s) for figure 7:

**Figure supplement 1.** The metabolic dependencies for IFNγ signaling are preserved in macrophages derived from myeloid progenitor lines.

---

electrons to an alternative acceptor. Additional work is necessary to elaborate the structure of the ETC in these cells and potential interactions with mTORC1.

Complex I is a metabolic hub with several core functions that cumulatively recycle nicotinamide adenine dinucleotide (NAD+), reduce ubiquinol, and initiate the PMF for ATP generation. While any or all of these physiological processes could contribute to IFNγ signaling, the differential effects of chemical inhibitors narrow the possibilities. Both rotenone and oligomycin inhibit the IFNγ response, and block electron flux through complex I, either directly or indirectly. In contrast, the ionophore CCCP disrupts the PMF and ATP generation without inhibiting electron transfer and does not affect IFNγ signaling. These observations indicate that the reduction state of the quinone pool and ATP generation does not regulate IFNγ responses in our system. Instead, our data indicate that complex I-dependent regeneration of NAD+ is the most likely regulator of IFNγ signaling. Indeed, NAD+ synthesis via either the de novo or salvage pathway is necessary for a variety of macrophage functions (*Cameron et al., 2019*; *Venter et al., 2014*; *Minhas et al., 2019*). Very recent work demonstrates an important role for NAD+ in STAT1 activation and PD-L1 induction by IFNγ in hepatocellular carcinoma cells (*Lv et al., 2021*). In this setting, inhibition of NAD+ synthesis reduces the abundance of phospho-STAT1 by disrupting a direct interaction with the ten-eleven translocation methylcytosine dioxygenase 1 (TET1). It remains unclear if a similar interaction occurs in the myeloid cells that are the focus of our work, as TET1 is expressed at very low levels in macrophages and splenic DCs (*Heng et al., 2008*). Regardless, these observations suggest that both NAD+ synthesis and its regeneration via mitochondrial respiration contribute to the IFNγ response in diverse cell types. This recently revealed interaction between metabolism and immunity could contribute to the observed association between NAD+ homeostasis and inflammatory diseases (*Minhas et al., 2019*), as well as the efficacy of checkpoint inhibitor therapy for cancer (*Lv et al., 2021*).

In the APC setting, we found that T cell activation required mitochondrial respiration. While complex I function, MHCII, and CD40 expression all largely correlate with T cell stimulation, our data indicate that additional IFNγ-inducible pathways also contribute to this activity. For example, unstimulated DCs basally express similar levels of MHCII as IFNγ-stimulated macrophages but are unable to productively present antigen to T cells. This observation suggests that additional aspects of antigen processing, presentation, or co-stimulation are IFNγ- and complex I-dependent. Similarly, MHC class I presentation machinery is transcriptionally induced upon IFNγ stimulation (*Rock et al., 2016*; *Van Rhijn et al., 2015*) and the induction of molecules recognized by donor-unrestricted T cells, such as MR1 and CD1, might also require additional signals to function. The specific effects of mitochondrial respiration on the type and quality of the T cell response will depend on how these diverse antigen-presenting and co-signaling molecules are influenced by cellular metabolic state.

The observation that IFNγ signaling depends on mitochondrial respiration provides a stark contrast to the well-established glycolytic dependency of many phagocyte functions, such as TLR signaling. This metabolic dichotomy between proinflammatory TLR signals and the IFNγ response mirrors known regulatory interactions between these pathways. For example, TLR stimulation has been shown to inhibit subsequent IFNγ responses via a number of target gene-specific mechanisms (*Benson and*

*Ernst, 2009*; *Fortune et al., 2004*; *Kincaid et al., 2007*; *Su et al., 2020*; *Jang and Javadov, 2020*). However, TLR stimulation also results in the disassembly of the ETC (*Su et al., 2020*; *Jang and Javadov, 2020*), which our observations predict to inhibit STAT1 phosphorylation and IFNγ signaling at the level of the receptor complex. More generally, our work suggests that fundamental metabolic programs contribute to the integration of activation signals by APC and influence the ultimate priming of an immune response.

# Materials and methods

## Key resources table

| Reagent type (species) or resource | Designation | Source or reference | Identifiers | Additional information |
|---|---|---|---|---|
| Cell line (*Mus musculus*) | L3-Cas9+ | Kiritsy and Ankley et al. (co-submitted) | | Primary BMDMs immortalized with J2 virus were transduced with Cas9 and single cell cloned |
| Cell line (*M. musculus*) | EGFP-Cas9 iBMDMs | This paper | | Primary BMDMs from Jackson Stock 026179 were immortalized with J2 virus |
| Cell line (*M. musculus*) | sgNdufa1 EGFP-Cas9 iBMDMs | This paper | | Cas9+ iBMDMs were transduced with Ndufa1 sgRNA |
| Cell line (*M. musculus*) | sgNdufa2 EGFP-Cas9 iBMDMs | This paper | | Cas9+ iBMDMs were transduced with Ndufa2 sgRNA |
| Cell line (*M. musculus*) | Cas9+ C57BL/6J estradiol-inducible HoxB8 progenitors | This paper | | Myeloid progenitors from Jackson stock 026179 were immortalized with HoxB8 retrovirus and maintained with 10 µM estradiol |
| Cell line (*M. musculus*) | sgNdufa1 C57BL/6J estradiol-inducible HoxB8 progenitors | This study | | Cas9+ HoxB8 cells were transduced with Ndufa1 sgRNA |
| Cell line (*M. musculus*) | Cas9+ C57BL/6J estradiol-inducible HoxB8 progenitors | This paper | | Myeloid progenitors from Jackson stock 003288 were immortalized with HoxB8 retrovirus and maintained with 10 µM estradiol |
| Strain, strain background (*M. musculus*) | C57BL6J | Jackson Laboratories | Stock 000664 | |
| Strain, strain background (*M. musculus*) | C7 TCR-transgenic mice (specific for ESAT-6 antigen) | PMID:18779346 | | Mice were donated to and maintained by the Behar laboratory |
| Peptide, recombinant protein | ESAT-6 peptide (MTEQQW NFAGIEAAA) | New England Peptide | | |
| Recombinant DNA reagent | sgOpti | Addgene | RRID:Addgene_ 85681 | |
| Recombinant DNA reagent | sgOpti with blasticidin and zeocyin selection | Kiritsy and Ankley et al. (co-submitted) | | sgOpti (RRID 85681) was modified with bacterial selection replaced with zeocyin and mammalian selection replaced with blasticidin |
| Recombinant DNA reagent | sgOpti with hygromycin and kanamycin selection | Kiritsy and Ankley et al. (co-submitted) | | sgOpti (RRID 85681) was modified with bacterial selection replaced with kanamycin and mammalian selection replaced with hygromycin |
| Recombinant DNA reagent | VSVG | Addgene | RRID:Addgene_8454 | |
| Recombinant DNA reagent | psPax2 | Addgene | RRID:Addgene_12260 | |
| Antibody | MHCII-PE, clone M5/114.15.2 (rat monoclonal) | BioLegend | RRID:AB_313323 | FC (1:800) |
| Recombinant DNA reagent | Mouse CRISPR KO pooled library (BRIE) | Addgene | RRID:Addgene_7363 | |
| Chemical compound, drug | Rotenone | Sigma | Cat# R8875 | |

*Continued on next page*

*Continued*

| Reagent type (species) or resource | Designation | Source or reference | Identifiers | Additional information |
|---|---|---|---|---|
| Chemical compound, drug | Oligomycin | Cayman | Cat# 11342 | |
| Chemical compound, drug | CCCP | Cayman | Cat# 25458 | |
| Chemical compound, drug | 1400W | Cayman | Cat# 81520 | |
| Chemical compound, drug | N-Acetyl cysteine | Cayman | Cat# 20261 | |
| Chemical compound, drug | Dimethyloxallyl glycine (DMOG) | Cayman | Cat# 71210 | |
| Chemical compound, drug | UK5099 | Cayman | Cat# 16,980 | |
| Chemical compound, drug | 2-Deoxyglucose (2-DG) | Cayman | Cat# 14325 | |
| Chemical compound, drug | MitoTEMPO hydrate | Cayman | Cat# 16621 | |
| Chemical compound, drug | Sodium azide | Sigma | Cat# S2002 | |
| Chemical compound, drug | Antimycin A | Sigma | Cat# A8674 | |
| Chemical compound, drug | Pam3CSK4 | Invivogen | Cat# tlrl-pms | |
| Chemical compound, drug | Linezolid (LZD) | Gift from Clifton Barry (used here PMID:32477361) | | |
| Antibody | Purified anti-STAT1 antibody clone A15158C (mouse monoclonal) | BioLegend | Cat# 603701; RRID:AB_2749867 | WB (1:1000) |
| Antibody | Phospho-Stat1 Tyr701, clone 58D6 (rabbit monoclonal) | Cell Signaling Technology | Cat# 5375; RRID:AB_10860071 | WB (1:1000) |
| Antibody | Purified anti-STAT1 Phospho Ser727 antibody, clone A15158B (mouse monoclonal) | BioLegend | Cat# 686408; RRID:AB_2650782 | WB (1:1000) |
| Antibody | Jak2 XP, clone D2E12 (rabbit monoclonal) | Cell Signaling Technology | Cat# 4040; RRID:AB_10691469 | WB (1:500) |
| Antibody | Phospho-Jak2 (Tyr1007/1008) antibody (unknown) | Cell Signaling Technology | Cat# 3771; RRID:AB_330403 | WB (1:500) |
| Antibody | Biotin anti-mouse CD119, IFNγ Rα chain antibody clone 2E2 (Armenian hamster monoclonal) | BioLegend | Cat# 112803; RRID:AB_2123476 | WB (1:1000) |
| Antibody | Anti-mouse β-actin antibody, clone C4 (mouse monoclonal) | Santa Cruz Biotechnology | Cat# sc-51850; RRID:AB_629337 | WB (1:2000) |
| Antibody | CD274-Bv421 clone 10F.9G2 (rat monoclonal) | BioLegend | RRID:AB_10897097 | FC (1:400) |
| Commercial assay or kit | Zombie Aqua Fixable Viability Kit | BioLegend | Cat# 423101 | FC (1:100) |
| Commercial assay or kit | DNeasy Blood and Tissue Kit | Qiagen | Cat# 69504 | |
| Antibody | CD40 APC anti-mouse CD40 antibody, clone 3/23 (rat monoclonal) | BioLegend | Cat# 124611; RRID:AB_1134081 | FC (1:200) |

*Continued on next page*

*Continued*

| Reagent type (species) or resource | Designation | Source or reference | Identifiers | Additional information |
|---|---|---|---|---|
| Antibody | Human anti-CD54, clone HCD54 (mouse monoclonal) | BioLegend | Cat# 322718; RRID:AB_2248731 | FC (1:400) |
| Antibody | Human anti-CD40 clone 5C3 (mouse monoclonal) | BioLegend | Cat# 334307; RRID:AB_1186060 | FC (1:400) |
| Antibody | Human anti-CD274, B7-H1, PD-L1, clone 29E (mouse monoclonal) | BioLegend | Cat# 329713; RRID:AB_10901164 | FC (1:400) |
| Antibody | Human anti-HLA-DR antibody, clone L243 (mouse monoclonal) | BioLegend | Cat# 307657; RRID:AB_2572100 | FC (1:400) |
| Antibody | Anti-mouse IFNγ antibody (rat monoclonal) | BioLegend | Cat# 505807; RRID:AB_315401 | FC (1:200) |
| Peptide, recombinant protein | Murine IL-12 | PeproTech | Cat# 210-12 | |
| Peptide, recombinant protein | Human GM-CSF | PeproTech | Cat# 300-03 | |
| Peptide, recombinant protein | Human IFN gamma | PeproTech | Cat# 300-02 | |
| Peptide, recombinant protein | Murine TNF | PeproTech | Cat# 315-01A | |
| Antibody | Anti-IL4 clone: 11B11 (rat monoclonal) | BioLegend | RRID:AB_2750407 | Neutralization (1:500) |
| Antibody | Goat anti-rabbit HRP (goat polyclonal) | Invitrogen | Cat# 31460 | WB (1:1000) |
| Antibody | Goat anti-mouse HRP (goat polyclonal) | Invitrogen | Cat# 31430 | WB (1:1000) |
| Commercial assay or kit | One-Step RT PCR Kit | Qiagen | Cat# 210215 | |
| Commercial assay or kit | Luna Universal One-Step RT-qPCR Kit | NEB | Cat# E3005 | |
| Commercial assay or kit | Trizol | Thermo Fisher Scientific | Cat# 15596026 | |
| Commercial assay or kit | CellTiter-Glo 2.0 Cell Viability Assay | Promega | Cat# G9241 | |
| Commercial assay or kit | Seahorse assay media | Agilent | Cat# 103575-100 | |
| Software, algorithm | MAGECK | PMID:25476604 | | |
| Peptide, recombinant protein | Interferon gamma | BioLegend | Cat# 575308 | |
| Commercial assay or kit | MojoSORT – Mouse CD4 Naïve T cell Isolation Kit | BioLegend | Cat# 480040 | |
| Commercial assay or kit | MojoSort Human CD14 Nanobeads | BioLegend | Cat# BioLegend 480093 | |
| Commercial assay or kit | IL6 ELISA-max | BioLegend | Cat# 431301 | |
| Commercial assay or kit | TNF ELISA-max | BioLegend | Cat# 430901 | |

*Continued on next page*

*Continued*

| Reagent type (species) or resource | Designation | Source or reference | Identifiers | Additional information |
|---|---|---|---|---|
| Commercial assay or kit | Human IL1b | R&D Systems | Cat# DY201 | |
| Commercial assay or kit | Human TNF-alpha DuoSet ELISA | R&D Systems | Cat# DY210 | |
| Commercial assay or kit | Greiss reagent | Promega | G2930 | |
| Sequence-based reagent | All oligonucleotide sequences are contained in **Supplementary file 1**. | This paper | | All oligonucleotide sequences are contained in **Supplementary file 1** |

## Cell culture

Cells were cultured in Dulbecco's Modified Eagle Medium (Gibco 11965118) supplemented with 10% fetal bovine serum (Sigma F4135), sodium pyruvate (Gibco 11360119), and HEPES (15630080). Primary BMDMs were generated by culturing bone marrow in the presence of media supplemented with 20% L929 supernatant for 7 days.

Immortalized macrophage cell lines in C57B/6J and Cas9-EGFP were established in using J2 retrovirus from supernatant of CREJ2 cells as previously described (*Blasi et al., 1989*). Briefly, isolated bone marrow was cultured in the presence of media enriched with 20% L929 supernatant. On day 3, cells were transduced with virus and cultured with virus for 2 days. Over the next 8 weeks, L929 media were gradually reduced to establish growth factor independence.

Conditionally immortalized myeloid progenitor cell lines were generated by retroviral transduction using an estrogen-dependent Hoxb8 transgene as previously described (*Wang et al., 2006*). Briefly, mononuclear cells were purified from murine bone marrow using Ficoll-Paque PLUS (GE Healthcare 17144002) and cultured in RPMI (Gibco 11875119) containing 10% fetal bovine serum (Sigma F4135), sodium pyruvate (Gibco 11360119), and HEPES (15630080), IL-6 (10 ng/mL; PeproTech #216-16), IL-3 (10 ng/mL; PeproTech #213-13), and SCF (10 ng/mL; PeproTech #250-03) for 48 hr. Non-adherent bone marrow cells from C57Bl/6J (Jax 000664), Cas9-EGFP knockin (Jax 026179), or Ifngr1 KO (Jax 003288) mice were transduced with ER-Hoxb8 retrovirus. After transduction, cells were cultured in media supplemented with supernatant from B16 cells expressing GM-CSF and 10 µM estradiol (Sigma E8875) to generate macrophage progenitor cell lines or in media supplemented with supernatant from B16 cells expressing FLT3L and 10 µM estradiol (Sigma E8875) to generate DC progenitor lines. To differentiate macrophages, progenitors were harvested and washed twice with PBS to remove residual estradiol and cultured in L929 supplemented media as above. To differentiate DCs (*Redecke et al., 2013*), progenitors were harvested, washed 2× with PBS, and cultured in FLT3-enriched complete RPMI for 8–10 days.

Human MDMs were differentiated from mononuclear cells of healthy donors. Briefly, peripheral blood mononuclear cells (PBMCs) were isolated from whole blood using Ficoll-Paque PLUS (GE Healthcare 17144002). CD14+ monocytes were purified using MojoSort Human CD14 Nanobeads (BioLegend 480093) according to the manufacturer's protocol. Cells were cultured in RPMI with 10% FBS, sodium pyruvate, and HEPES and supplemented with recombinant GM-CSF (50 ng/mL, PeproTech 300-03) for 6 days. Thaws were harvested using Accutase (Gibco A1110501).

## Cell stimulations

Murine IFNγ (PeproTech 315-05) and human IFNγ (PeproTech 300-02) were used at 10 ng/mL unless indicated otherwise in the figure legends. Murine TNF (315-01A) was used at 25 ng/mL. Pam3CSK4 (Invivogen tlrl-pms) was used at 200 ng/mL.

## CRISPR screens

A clonal macrophage cell line stably expressing Cas9 (L3) was established as described elsewhere (*Kiritsy et al., 2021*). A plasmid library of sgRNAs targeting all protein coding genes in the mouse genome (Brie Knockout library, Addgene 73633) was packaged into lentivirus using HEK293T cells. HEK293T supernatants were collected and clarified, and virus was titered by quantitative real-time PCR and colony counting after transduction of NIH3T3. L3 cells were transduced at a multiplicity

of infection (MOI) of ~0.2 and selected with puromycin 48 hr after transduction (2.5  μg/mL). The library was minimally expanded to avoid skewing mutant representation and then frozen in aliquots in freezing media (90%  FBS, 10%  DMSO).

Two replicate screens for MHCII, CD40, and PD-L1 were performed as follows: 2e8 cells of the KO library were stimulated with IFNγ (10 ng/mL; PeproTech 315-05) for 24 hr after which cells were harvested by scraping to ensure integrity of cell surface proteins. Cell were stained with TruStain FcX anti-mouse CD16/32 (BioLegend 101319) and LIVE/DEAD Fixable Aqua (Invitrogen L34957) as per the manufacturer's instructions. For each of the respective screens, stimulated library was stained for its respective marker with the following antibody: MHCII (APC anti-mouse I-A/I-E Antibody, clone M5/114.15.2, BioLegend 107613), CD40 (APC anti-mouse CD40 antibody, clone 3/23, BioLegend 124611), or PD-L1 (APC anti-mouse CD274 [B7-H1, PD-L1] antibody, clone 10F.9G2, BioLegend 124311). Each antibody was titrated for optimal staining using the isogenic L3 macrophage cell line. Following staining, cells were fixed in 4%  paraformaldehyde. High- and low-expressing populations were isolated by FACS using a BD FACS Aria II Cell Sorter. Bin size was guided by control cells that were unstimulated and to ensure sufficient library coverage (>25×  unselected library, or >2e6 cells per bin). Following isolation of sorted populations, paraformaldehyde crosslinks were reversed by incubation in proteinase K (Qiagen) at 55° for 6–8 hr. Subsequently, genomic DNA was isolated using DNeasy Blood and Tissue Kit (Qiagen 69504) according to the manufacturer's instructions. Amplification of sgRNAs by PCR was performed as previously described (*Doench et al., 2016*; *Joung et al., 2017*) using Illumina compatible primers from IDT, and amplicons were sequenced on an Illumina NextSeq500. Sequence reads were trimmed to remove adapter sequence and adjust for staggered forward (p5) primer using Cutadapt v2.9. Raw sgRNA counts for each sorted and unsorted (input library) population were quantified using bowtie2 via MAGeCK to map reads to the sgRNA library index (no mismatch allowed); an sgRNAindex was modified to reflect genes transcribed by our macrophage cell line either basally or upon stimulation with IFNγ as previously published (*Kiritsy et al., 2021*). Counts for sgRNAs were median normalized to account for variable sequencing depth.

## MAGeCK-MLE

We used MAGeCK-MLE to test for gene enrichment. Two separate analyses were performed in order to (1) identify regulators of the IFNγ response and (2) specific regulators of each of the screen targets. For both analyses, the baseline samples were the input libraries from each of the replicate screens in order to account for slight variabilities in library distribution for each screen. For (1), the generalized linear model was based on a design matrix that was 'marker-blind' and only considered the bin of origin (i.e., MHCII-low, CD40-low, PD-L1-low vs. MHCII-high, CD40-high, PD-L1-high). For (2), the design matrix was 'marker-aware and bin-specific' to test for marker-specific differences (i.e., MHCII-low vs. CD40-low vs. PD-L1-low); the analysis was performed separately for each bin, low- or high-expressing mutants, to identify marker-specific positive and negative regulators, respectively. For each analysis, β scores (selection coefficient) for each gene were summed across conditions to allow for simultaneous assessment of positive and negative regulators across conditions. Data are provided in *Supplementary file 1*.

GSEA was performed using a ranked gene list as calculated from MAGeCK-MLE beta scores and false discovery rate (FDR). To facilitate the identification of positively and negatively enriched gene sets from the high- and low-expressing populations, the positive ('pos | beta') and negative ('neg | beta') beta scores for each gene were summed as described above ('beta_sum'). To generate a ranked gene list for GSEA, we employed Stouffer's method to sum positive ('pos | z') and negative ('neg | z') selection z-scores, which were used to recalculate p-values ('p_sum') as has been previously described (*Brown, 1975*; *Jia et al., 2017*; *Bodapati et al., 2020*). Using these summative metrics, we calculated a gene score as log10(p_sum) * (beta_sum). Genes were ranked in descending order, and GSEA was performed with standard settings including 'weighted' enrichment statistic and 'meandiv' normalization mode. Analysis was inclusive of gene sets comprising 10–500 genes that were compiled and made available online by the Bader lab (*Merico et al., 2010*; *Reimand et al., 2019*).

## Plasmids and sgRNA cloning

Lentivirus was generated using HEK293T cells with packaging vector psPAX2 (Addgene #12260) and envelope plasmid encoding VSV-G. Transfections used TransIT-293 (MirusBio MIR 2704) and plasmid

ratios were calculated according to the manufacturer's instructions. For the generation of retrovirus, pCL-Eco in place of separate packaging and envelope plasmid was used. Retrovirus encoding the ER-Hoxb8 transgene was kindly provided by David Sykes. sgOpti was a gift from Eric Lander and David Sabatini (Addgene plasmid #85681; *Fulco et al., 2016*). Individual sgRNAs were cloned as previously described. Briefly, annealed oligos containing the sgRNA targeting sequence were phosphorylated and cloned into a dephosphorylated and BsmBI (New England Biolabs) digested SgOpti (Addgene #85681), which contains a modified sgRNA scaffold for improved sgRNA-Cas9 complexing. Use of sgOpti derivatives for delivery of multiple sgRNAs was performed as detailed elsewhere (*Kiritsy et al., 2021*). The sgRNA targeting sequences used for cloning were as follows:

| Name/target | sgRNA sequence |
| --- | --- |
| sgIfngr1_1 | TATGTGGAGCATAACCGGAG |
| sgIfngr1_2 | GGTATTCCCAGCATACGACA |
| sgIrf1_1 | CTGTAGGTTATACAGATCAG |
| sgIrf1_2 | CGGAGCTGGGCCATTCACAC |
| sgPtpn2_1 | AAGAAGTTACATCTTAACAC |
| sgPtpn2_2 | TGCAGTGATCCATTGCAGTG |
| sgNdufa1_1 | TGTACGCAGTGGACACCCCG |
| sgNdufa1_2 | CGCGTTCCATCAGATACCAC |
| sgNdufa2_1 | GCAGGGATTTCATCGTGCAA |
| sgNdufa2_2 | ATTCGCGGATCAGAATGGGC |
| sgStat1_1 | GGATAGACGCCCAGCCACTG |
| sgStat1_2 | TGTGATGTTAGATAAACAGA |
| sgOstc_1 | GCGTACACCGTCATAGCCGA |
| sgOstc_2 | TCTTACTTCCTCATTACCGG |
| sgCnbp_1 | AGGTAAAACCACCTCTGCCG |
| sgCnbp_2 | GTTGAAGCCTGCTATAACTG |

## Flow cytometry

Cells were harvested at the indicated times post-IFNγ stimulation by scrapping to ensure intact surface proteins. Cells were pelleted and washed with PBS before staining with TruStain FcX anti-mouse CD16/32 (BioLegend 101319) or TruStain FcX anti-human (BioLegend 422301) and LIVE/DEAD Fixable Aqua (Invitrogen L34957) as per the manufacturer's instructions. The following antibodies were used as indicated in the figure legends:

APC-Fire750 anti-mouse I-A/I-E antibody, clone M5/114.15.2, BioLegend 107651
PE anti-mouse CD40 antibody, clone 3/23, BioLegend 124609
Brilliant Violet 421 anti-mouse CD274 (B7-H1, PD-L1) antibody, clone 10F.9G2, BioLegend 124315
Alexa Fluor 647 anti-human CD54 antibody, clone HCD54, BioLegend 322718
PE anti-human CD40 antibody, clone 5C3, BioLegend 334307
Brilliant Violet 421 anti-human CD274 (B7-H1, PD-L1) antibody, clone 29E.2A3, BioLegend 329713
APC/Fire 750 anti-human HLA-DR antibody, clone L243, BioLegend 307657

For intracellular cytokine staining, cells were treated with brefeldin A (BioLegend 420601) for 5 hr before harvesting. Following staining and fixation, cells were permeabilized (BioLegend 421002) and stained according to the manufacturer's protocol using the following antibodies: PE anti-mouse IFNγ antibody, BioLegend 505807.

Surface protein expression was analyzed on either a MacsQuant Analyzer or Cytek Aurora. All flow cytometry analyses were done in FlowJo V10 (TreeStar).

## Chemical inhibitors

All chemical inhibitors were used for the duration of cell stimulation unless otherwise stated. Rotenone (Sigma R8875) was resuspended in DMSO and used at 10 µM unless indicated otherwise in the figure legends. Oligomycin (Cayman 11342) was resuspended in DMSO and used at 2.5 µM unless indicated otherwise. CCCP (Cayman 25458) was resuspended in DMSO and used at 1.5 µM unless indicated otherwise. 1400W hydrochloride (Cayman 81520) was resuspended in culture media, filter sterilized, and used immediately at 25 µM unless indicated otherwise. NAC (Cayman 20261) was resuspended in culture media, filter sterilized, and used immediately at 10 mM. DMOG (Cayman 71210) was resuspended in DMSO and used at 200 µM. UK5099 (Cayman 16980) was resuspended in DMSO and used at 20 µM. 2-Deoxy-D-glucose (2DG, Cayman 14325) was resuspended in culture media, filter sterilized, and used at 1 mM or at the indicated concentrations immediately. MitoTEMPO hydrate (Cayman 16621) was resuspended in DMSO and used at the indicated concentrations. Antimycin A (Sigma A8674) and sodium azide (Sigma S2002) were resuspended in DMSO, filter sterilized, and used at indicated concentrations.

For experiments that used defined minimal media with carbon supplementation, D-galactose, sodium pyruvate, and D-glucose were used at 10 mM in DMEM without any carbon (Gibco A1443001). For establishment of macrophage cell line with diminished mitochondrial mass, cells were continuously cultured in LZD (kind gift from Clifton Barry) for 4 weeks at 50 µg/mL or DMSO control. Both LZD-conditioned and DMSO control lines were supplemented with uridine at 50 µg/mL. Prior to experimentation, cells were washed with PBS and cultured without LZD for at least 12 hr.

## ELISA and nitric oxide quantification

The following kits were purchased from R&D Systems or BioLegend for quantifying protein for cell supernatants:

> Mouse IL-6 DuoSet ELISA (DY406) or BioLegend ELISA-max (431301)
> Mouse TNF-alpha DuoSet ELISA (DY410) or BioLegend ELISA-max (430901)
> Mouse IFN-gamma DuoSet ELISA (DY485)
> Human IL-1 beta/IL-1F2 DuoSet ELISA (DY201)
> Human TNF-alpha DuoSet ELISA (DY210)

Nitric oxide was quantified from cell supernatants using the Griess Reagent System according to the manufacturer's instructions (Promega G2930). For these experiments, cell culture media without phenol red (Gibco A1443001 or Gibco 31053028) were used.

## RNA isolation and quantitative real-time PCR

To isolate RNA, cells were lysed in TRIzol (15596026) according to the manufacturer's instructions. Chloroform was added to lysis at a ratio of 200 µL chloroform per 1 mL TRIzol and centrifuged at 12,000 × g for 20 min at 4°C. The aqueous layer was separated and added to equal volume of 100% ethanol. RNA was isolated using the Zymo Research Direct-zol RNA extraction kit. Quantity and purity of the RNA were checked using a NanoDrop and diluted to 5 ng/µL in nuclease-free water before use. Quantitative real-time PCR was performed using NEB Luna Universal One-Step RT-qPCR Kit (E3005) or the Quantitect SYBR green RT-PCR kit (204243) according to the manufacturer's protocol and run on a Viia7 thermocycler or StepOne Plus Theromocycler. Relative gene expression was determined with ddCT method with β-actin transcript as the reference.

| Primer | Sequence |
| --- | --- |
| RT_Actb-1F | GGCTGTATTCCCCTCCATCG |
| RT_Actb-1R | CCAGTTGGTAACAATGCCATGT |
| RT_Cd274-1F | GCTCCAAAGGACTTGTACGTG |
| RT_Cd274-1R | TGATCTGAAGGGCAGCATTTC |
| RT-Ciita-1F | AGACCTGGATCGTCTCGT |
| RT-Ciita-1R | AGTGCATGATTTGAGCGTCTC |

*Continued on next page*

*Continued*

| Primer | Sequence |
|---|---|
| RT-Gapdh-1F | TGGCCTTCCGTGTTCCTAC |
| RT-Gapdh-1R | GAGTTGCTGTTGAAGTCGCA |

## Quantification of mitochondrial genomes

Genomic DNA was isolated from cell pellets using the DNeasy Blood and Tissue Kit (Qiagen 69504). Quantitative PCR was run using NEB Luna Universal One-Step RT-qPCR without the RT enzyme mix and run on a Viia7 thermocycler. Relative quantification of mitochondrial genomes was determined by measuring the relative abundance of mitochondrially encoded gene Nd1 to the abundance of nuclear encoded Hk2 as has been described elsewhere (*Field et al., 2020*). All primers are detailed as follows:

| Name/target | Sequence |
|---|---|
| Mm-Nd1-1F | CTAGCAGAAACAAACCGGGC |
| Mm-Nd1-1R | CCGGCTGCGTATTCTACGTT |
| Mm-Hk2-1F | GCCAGCCTCTCCTGATTTTAGTGT |
| Mm-Hk2-1R | GGGAACACAAAAGACCTCTTCTGG |

## Immunoblot

At the indicated times following stimulation, cells were washed with PBS once and lysed in ice using the following buffer: 1% Triton X-100, 150 mM NaCl, 5 mM KCl, 2 mM $MgCl_2$, 1 mM EDTA, 0.1% SDS, 0.5% DOC, 25 mM Tris-HCl, pH 7.4, with protease and phosphatase inhibitor (Sigma #11873580001 and Sigma P5726). Lysates were further homogenized using a 25 g needle and cleared by centrifugation before quantification (Pierce BCA Protein Assay Kit, 23225). Parallel blots were run with the same samples, 15 µg per well. The following antibodies were used according to the manufacturer's instructions:

> Purified anti-STAT1 antibody, BioLegend, clone A15158C
> Purified anti-STAT1 phospho (Ser727) antibody, BioLegend, clone A15158B
> Phospho-Stat1 (Tyr701) rabbit mAb, Cell Signaling Technology, clone 58D6
> Jak2 XP rabbit mAb, Cell Signaling Technology, clone D2E12
> Phospho-Jak2 (Tyr1007/1008) antibody, Cell Signaling Technology, #3771S
> Anti-mouse β-actin antibody, Santa Cruz Biotechnology, clone C4
> Biotin anti-mouse CD119 (IFN-γ Rα chain) antibody, BioLegend, clone 2E2
> Goat anti-rabbit IgG (H + L) secondary antibody, HRP, Invitrogen 31460
> Goat anti-mouse IgG (H + L) secondary antibody, HRP, Invitrogen 31430
> HRP-conjugated streptavidin, Thermo Scientific N100.

## Bioenergetics assays

Relative glycolytic and respiratory capacities were determined as has previously been demonstrated (*Horlbeck et al., 2018*). Briefly, cellular ATP levels were determined using CellTiter-Glo 2.0 Cell Viability Assay (Promega G9241) according to the manufacturer's protocol. Cells were grown in the conditions indicated in the figure legends for 4 hr unless stated otherwise. ATP levels were normalized according to the figure legends.

## Seahorse metabolic rate assays with BMDMs

Wild-type (C57Bl/6J) BMDMs were seeded in a Seahorse cell culture plate at $10^5$ cells per well and stimulated for 24 hr with recombinant murine IFNγ (10 ng/mL) or Pam3CSK4 (200 ng/mL). Cellular oxidative phosphorylation and glycolysis were measured using the Seahorse Bioscience Extracellular Flux Analyzer (XFe96, Seahorse Bioscience Inc, North Billerica, MA) by measuring OCR (indicative of respiration) and ECAR (indicative of glycolysis) in real time according to the manufacturer's protocol. Prior to measurements, culture media were removed and replaced with 180 µL pH ready Seahorse

Assay Media (Agilent; Catal#103575-100) and incubated in the absence of $CO_2$ for 1 hr in the BioTek Cytation1 instrument during which time pre-assay brightfield images were collected. Basal levels of OCR and ECAR were recorded, then OCR and ECAR levels following injection of compounds that inhibit the mitochondrial ETC, or ATP synthesis were monitored. As per the manufacturer's protocol for the Mito Stress Test, assay cells were sequentially treated with oligomycin (2 µM), carbonyl cyanide-4-(trifluoromethoxy)phenylhydrazone (FCCP) (0.5 µM), and rotenone + antimycin A (0.5 µM). OCR and ECAR were then measured in a standard 6 min cycle of mix (2 min), wait (2 min), and measure (2 min). All OCR and ECAR values were normalized following the Seahorse normalization protocol. Briefly after, the assay cells were stained with 2 µg/mL Hoechst 33342 (Thermo Fisher Scientific) for 30 min while performing post-assay brightfield imaging. Cells were then imaged and counted using the BioTek Cytation1. Cell counts were calculated by Cell Imaging software (Agilent) and imported into Wave (Agilent) using the normalization function.

## T cell activation assay

We used a previously established co-culture system to assess antigen presentation to Ag-specific T cells. Briefly, C7 CD4+ T cells were isolated from transgenic C7 mice, respectively, and stimulated in vitro with irradiated splenocytes pulsed with the ESAT-61-15 peptide in complete media (RPMI with 10% FBS) containing IL-2 and IL-7. After the initial stimulation, the T cells were split every 2 days for 3–4 divisions and rested for 2–3 weeks. After the initial stimulation, the cells were cultured in complete media containing IL-2 and IL-7. The following synthetic peptide epitopes were used as antigens from New England Peptide (Gardener, MA): ESAT-61-15 (MTEQQWNFAGIEAAA).

For use in co-culture assay, T cells were added to peptide-pulsed macrophages as described in the figure legends at an effector to target ratio of 1:1. Following 1 hr of co-culture, brefeldin A was added for 5 hr before assessing intracellular cytokine production by ICS.

## Quantification of subunit effects on N-module

We used publicly available proteomics data in which the protein abundance of all complex I subunit was measured when each subunit was genetically deleted (*Stroud et al., 2016*). As determined empirically by the authors, the N-module components included NDUFA1, NDUFA2, NDUFS1, NDUFV2, NDUFA6, NDUFS6, NDUFA7, NDUFS4, and NDUFV3. The relative effect of each subunit (using a KO of that subunit) on N-module protein stability was calculated as the sum of the median log2 ratio of each of the abovementioned subunits minus the median log2 ratio of itself (since it is knocked out).

## Statistical analysis, replicates, grouping, and figures

Statistical analysis was done using Prism version 7 (GraphPad) as indicated in the figure legends. Data are presented, unless indicated otherwise, as the mean ± standard deviation. Throughout the article, no explicit power analysis was used, but group size was based on previous studies using similar approaches. Throughout the article, biological replicate refers to independent wells or experiments processed at similar times. For RT-PCR experiments, technical replicates were used and are defined as repeat measures from the same well. Throughout the article, groups were assigned based on genotypes and blinding was not used throughout. Independent personnel completed several key figures to ensure robustness. Figures were created in Prism V8, R (version 3.6.2). MAGeCK-MLE was used as part of MAGeCK-FLUTE package v1.8.0 or was created with BioRender.com.

## Acknowledgements

We thank all the members of the Sassetti, Behar, and Olive labs for critical feedback and input throughout the project. A special thanks to Megan K Proulx, Mario Meza Segura, and the donors for their assistance and expertise to the human macrophage derivation We thank the flow cytometry core at UMMS for their help in all experiments requiring flow cytometry. This work was supported by startup funding to AJO provided by Michigan State University, support from the Arnold O Beckman Postdoctoral fellowship to AJO, and grants from the NIH (AI146504, AI132130), DOD (W81XWH2010147), and USDA (NIFA HATCH 1019371).

## Additional information

### Funding

| Funder | Grant reference number | Author |
|---|---|---|
| National Institutes of Health | AI146504 | Andrew J Olive |
| National Institutes of Health | AI132130 | Christopher M Sassetti |
| U.S. Department of Defense | W81XWH2010147 | Andrew J Olive |
| U.S. Department of Agriculture | NIFA HATCH 1019371 | Andrew J Olive |

The funders had no role in study design, data collection and interpretation, or the decision to submit the work for publication.

### Author contributions

Michael C Kiritsy, Conceptualization, Investigation, Methodology, Validation, Visualization, Writing - original draft, Writing - review and editing; Katelyn McCann, Formal analysis, Investigation, Writing - review and editing; Daniel Mott, Investigation, Methodology, Writing - review and editing; Steven M Holland, Resources, Supervision; Samuel M Behar, Methodology, Supervision, Writing - review and editing; Christopher M Sassetti, Conceptualization, Funding acquisition, Supervision, Writing - original draft, Writing - review and editing; Andrew J Olive, Conceptualization, Formal analysis, Funding acquisition, Investigation, Methodology, Supervision, Validation, Writing - original draft, Writing - review and editing

### Author ORCIDs

Michael C Kiritsy http://orcid.org/0000-0001-8364-8088
Samuel M Behar http://orcid.org/0000-0002-3374-6699
Christopher M Sassetti http://orcid.org/0000-0001-6178-4329
Andrew J Olive http://orcid.org/0000-0003-3441-3113

### Ethics

All human blood samples were donated following informed consent and approved under IRB protocol I-375-19.

This study was performed in strict accordance with the recommendations in the Guide for the Care and Use of Laboratory Animals of the National Institutes of Health. All of the animals were handled according to approved institutional animal care and use committee (IACUC) protocols (PROTO201800057) of Michigan State University and (Protocol# 1649) of the University of Massachusetts Medical School.

### Decision letter and Author response

Decision letter https://doi.org/10.7554/eLife.65109.sa1
Author response https://doi.org/10.7554/eLife.65109.sa2

## Additional files

### Supplementary files
• Supplementary file 1. Oligonucleotides used in this study.
• Transparent reporting form

### Data availability

Raw sequencing data in FASTQ and processed formats is available for download from NCBI Gene Expression Omnibus (GEO) under accession number GSE162463.

The following dataset was generated:

| Author(s) | Year | Dataset title | Dataset URL | Database and Identifier |
|---|---|---|---|---|
| Kiritsy MC, Sassetti CM, Olive AJ | 2020 | Mitochondrial respiration contributes to the interferon gamma response in antigen presenting cells | https://www.ncbi.nlm.nih.gov/geo/query/acc.cgi?acc=GSE162463 | NCBI Gene Expression Omnibus, GSE162463 |

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
