## [Editor Report]

In this article, Olive and colleagues used a genetic screen to identify complex I (CI) of the electron transport chain (ETC) as a regulator of IFNγ-mediated gene expression in macrophages. They attribute this role of CI to effects on the activity of the JAK-STAT pathway downstream of the IFNγ receptor. That CI (or perhaps ETC) activity can acutely regulate JAK-STAT signaling has interesting implications for the metabolic regulation of signal transduction, and the underpinning basis would be important to elucidate in future studies.

---

## [Decision Letter]

**Decision letter after peer review:**

Thank you for submitting your article "Mitochondrial respiration contributes to the interferon γ response in antigen presenting cells" for consideration by *eLife*. Your article has been reviewed by 2 peer reviewers, one of whom is a member of our Board of Reviewing Editors, and the evaluation has been overseen by Wendy Garrett as the Senior Editor. The following individual involved in review of your submission has agreed to reveal their identity: Ivan Zanoni (Reviewer #2).

The reviewers have discussed the reviews with one another and the Reviewing Editor has drafted this decision to help you prepare a revised submission.

As the editors have judged that your manuscript is of interest, but as described below that additional experiments are required before it is published, we would like to draw your attention to changes in our revision policy that we have made in response to COVID-19 (https://elifesciences.org/articles/57162). First, because many researchers have temporarily lost access to the labs, we will give authors as much time as they need to submit revised manuscripts. We are also offering, if you choose, to post the manuscript to bioRxiv (if it is not already there) along with this decision letter and a formal designation that the manuscript is "in revision at eLife". Please let us know if you would like to pursue this option. (If your work is more suitable for medRxiv, you will need to post the preprint yourself, as the mechanisms for us to do so are still in development.)

Summary:

In this manuscript, Olive and colleagues used a genetic screen to identify Complex I (CI) of the electron transport chain (ETC) as a regulator of IFNγ-mediated gene expression in macrophages. They attribute this role of CI to effects on the activity of the JAK-STAT pathway downstream of the IFNγ receptor.

While a potential link between CI activity and the activity of the JAK-STAT pathway would be interesting, the reviewers think that additional analyses are needed to substantiate this claim and rule out alternative interpretations.

Essential revisions:

1. Lines 204-205: The authors find that sgRNAs targeting other complexes of the ETC, including CIII and CIV, had no effect on the ability of IFNγ to stimulate expression of cell surface markers. How do the authors interpret these findings, since CI does not work in isolation in the ETC and is rather dependent on CIII and CIV activity?

2. How does IFNγ stimulation affect oxidative metabolism as assessed by Seahorse? In order to corroborate the authors' conclusions regarding activity of individual ETC complexes (point 1 above), Seahorse analysis of individual complexes is also advised.

3. The authors do some limited analyses in human MDMs to suggest that their findings in the mouse macrophage cell line can be generalized to other macrophage populations. It would be great if the analyses in the human MDMs could be extended to further strengthen the generality of their central findings.

4. Figure 6D: Not clear whether similar exposures were used in different panels. Would be better to load samples in the same gel so that the same exposure can be used and a direct comparison between conditions can be made.

5. Figure 6D: Does acute treatment with rotenone (but not inhibitors of other ETC complexes) have similar effects in reducing JAK-STAT signaling as knockdown of CI subunits? If not, then stable, long-term knockdown of CI subunits may have some effect independent of respiration in influencing JAK-STAT signaling (for example, on expression of some component of the JAK-STAT pathway). This interpretation could also explain why knockdown of other components of the ETC do not have similar effects to CI. Rotenone treatment could be tried (and compared with inhibitors of other ETC complexes), and if the data are different from knockdown of CI subunits, then related data in the study could be re-interpreted and conclusions modified.

6. In Figure 3H a key control is missing. What about survival of the cells when the import of the only energy substrate is blocked?

7. The authors could consider placing their findings in the context of the broader literature. (As just one example, Ivashkiv Nat Imm 2015 described a role for mTORC1 and metabolism in IFNγ-mediated transcriptional and translational regulation in macrophages.) This would increase the impact of their findings.

*Reviewer #1:*

In this manuscript, Olive and colleagues used a genetic screen to identify mitochondrial Complex I as a regulator of IFNγ-mediated gene expression in macrophages. They attribute this role of CI to effects on the activity of the JAK-STAT pathway downstream of the IFNγ receptor. They show that T cell stimulatory activity is a consequence of CI activity.

There are some questions regarding the authors' proposed mechanism for how CI activity influences IFN-mediated gene expression in macrophages:

1. Lines 204-205: The authors find that sgRNAs targeting other complexes of the ETC, including CIII and CIV, had no effect on the ability of IFNγ to stimulate expression of cell surface markers. How do the authors interpret these findings, since CI does not work in isolation in the ETC and is rather dependent on CIII and CIV activity?

2. How does IFNγ stimulation affect ETC activity, for example as assessed by Seahorse?

3. Figure 6D: Not clear whether similar exposures were used in different panels. Would be better to load samples in the same gel so that the same exposure can be used and a direct comparison between conditions can be made.

4. Figure 6D: Does acute treatment with rotenone have similar effects in reducing JAK-STAT signaling as knockdown of CI subunits? If not, then stable, long-term knockdown of CI subunits may have some effect independent of respiration in influencing JAK-STAT signaling (for example, on expression of some component of the JAK-STAT pathway). This interpretation could also explain why knockdown of other components of the ETC do not have similar effects to CI. Rotenone treatment could be tried and if the data are different from knockdown of CI subunits, then related data in the study could be re-interpreted and conclusions modified.

5. The authors propose that "CI-dependent regeneration of NAD is the most likely regulator of IFNγ signaling". This conclusion is based on their observations that oligomycin but not CCCP has a similar effect to rotenone, but many other interpretations are possible. To suggest NAD regeneration as being causal seems a bit of stretch unless some supporting data is provided. For example, does addition of NAD precursors (e.g., NAM) rescue the defects in the CI subunit knockdown?

Other concerns:

1. The authors do not mention which macrophages (primary, cell line) they use for their screen.

*Reviewer #2:*

– In figure 1E the authors suggest the importance of TNF receptor in regulating CD40 expression. Did the authors measure TNF production in the supernatant of their cell line treated with IFNγ? Based on the literature and on Figure 5C, there should be no TNF in the culture of IFNγ-treated macrophages. If so, how is it possible that the TNFR is involved? The authors should at least try to block TNF and/or TNFR with blocking antibodies to assess if suboptimal doses of TNF are playing any role.

– Throughout the paper: the authors often "measure" respiration and/or OXPHOS via measuring ATP levels (eg. Figure 3B and many others). The paper would highly benefit from the use of a seahorse to determine how the treatments made throughout the manuscript affect oxygen consumption and/or the extracellular acidification rate. Using ad hoc methods, it is possible not only to specifically measure OXPHOS and glycolysis, but also to ask important questions when substrates, such as pyruvate or citrate, are used. Most of the conclusions driven on the relevance of either respiration or OXPHOS, as well as ATP production sources, are very indirect and a seahorse would be required to really prove that the conclusions made are correct.

– The introduction of the cytokine measurement in figure 4 and 5 is indeed important but quite misleading. No real comparison with the findings published by Ivashkiv's group in Nature Immunology in 2015 is made. This paper looks at the involvement of mTORC1, metabolism and other kinases in the transcriptional and translational regulation driven by IFNγ in macrophages. mTORC1 is a key regulator of glycolysis, and IFNγ is shown to inhibit mTORC1, if so, one would expect respiration and/or other metabolic changes may occur and may be somewhat linked to complex I. Also, the fact that in the absence/block of complex I, cytokines in cells treated with TLR2 agonists are boosted is very confounding, and it makes hard to interpret data when complex I is blocked and IFNγ is also administered together with TLR2 stimulation. The authors should either better explain these phenotypes, or only focus on signals that do not involve TLR2 stimulation, leaving these data for a future work.

– In Figure 3H a key control is missing. What about survival of the cells when the import of the only energy substrate is blocked?

– The experiment with CCCP suggests that Complex I may regulate macrophage functions independently of OXPHOS. This is maybe the most exciting point of the paper, and it is partially addressed in figure 6D where phosphorylation of JAKs and STATs is analyzed. How do we go from Complex I to the most proximal signaling components of the IFNGR?

---

## [Author Response]

Essential revisions:1. Lines 204-205: The authors find that sgRNAs targeting other complexes of the ETC, including CIII and CIV, had no effect on the ability of IFNγ to stimulate expression of cell surface markers. How do the authors interpret these findings, since CI does not work in isolation in the ETC and is rather dependent on CIII and CIV activity?

This is a very interesting question, and we agree that it was not addressed directly enough in the original manuscript. Based on the data in the manuscript, it remained possible that CIII and CIV were not identified in the screen due to a technical issue, such as gene essentiality. Alternatively, these data could suggest that CI is still able to support IFNγ signaling in the absence of these downstream complexes, perhaps by transferring electrons to an alternative acceptor. In the revised manuscript, we test this directly using small molecule inhibition of CIII and CIV using antimycin A and sodium azide respectively. While treatment with either inhibitor reduced ATP to a similar degree as rotenone, the effect on IFNγ-mediated MHCII expression was significantly less. These results are very similar to what we observed with oligomycin which reduced the IFNγ responses but not to the same magnitude as complex I inhibition. Both the data from our screen and these more direct inhibition data support a preferential role complex I in the IFNγ response. While a detailed investigation electron flux and alternative acceptors is clearly outside the scope of this manuscript, we dedicate a new section in the discussion to this topic.

2. How does IFNγ stimulation affect oxidative metabolism as assessed by Seahorse? In order to corroborate the authors' conclusions regarding activity of individual ETC complexes (point 1 above), Seahorse analysis of individual complexes is also advised.

As suggested, we have performed Seahorse analysis on IFNγ- treated macrophages. The new Figure 2F shows that IFNγ increases basal ECAR and OCR of macrophages. This panel also incorporates inhibitors of individual complexes, showing that both basal and maximal OCR are increased following IFNγ. These data are now discussed in lines 233-241.

3. The authors do some limited analyses in human MDMs to suggest that their findings in the mouse macrophage cell line can be generalized to other macrophage populations. It would be great if the analyses in the human MDMs could be extended to further strengthen the generality of their central findings.

This was an excellent suggestion. In the new Supplemental Figure 2, we now provide the requested data from monocytes and M-CSF derived macrophages, which are consistent with the GM-CSF-derived MDM studies in the original manuscript (current Figure 5). Thus, the role for OXPHOS in IFNγ responses are common to a variety of human monocyte/macrophage cell types.

4. Figure 6D: Not clear whether similar exposures were used in different panels. Would be better to load samples in the same gel so that the same exposure can be used and a direct comparison between conditions can be made.

This entire experiment was redone. In addition to loading all samples on the same gel, we have included a rotenone treatment, as suggested in Point 5. These studies unequivocally show that chemical or genetic inhibition of complex I reduces IFNγ receptor signaling.

5. Figure 6D: Does acute treatment with rotenone (but not inhibitors of other ETC complexes) have similar effects in reducing JAK-STAT signaling as knockdown of CI subunits? If not, then stable, long-term knockdown of CI subunits may have some effect independent of respiration in influencing JAK-STAT signaling (for example, on expression of some component of the JAK-STAT pathway). This interpretation could also explain why knockdown of other components of the ETC do not have similar effects to CI. Rotenone treatment could be tried (and compared with inhibitors of other ETC complexes), and if the data are different from knockdown of CI subunits, then related data in the study could be re-interpreted and conclusions modified.

This was a very insightful comment. As discussed above, this suggested experiment was performed and we found that short-term rotenone treatment reduces Stat1 and Jak2 phosphorylation to a similar degree as genetic inactivation of this complex (Figure 6D). This experiment strengthens our conclusion that complex I activity is necessary for the IFNγ response.

6. In Figure 3H a key control is missing. What about survival of the cells when the import of the only energy substrate is blocked?

This comment helped us realize that these data were presented in a confusing manner. The reviewer is asking if blockade of pyruvate transport results in increased cell death when pyruvate is the sole carbon source. We now realize that including this piece of data is needlessly confusing. The purpose of this experiment is to show that the IFNγ response is intact when the glucose in the medium was replaced with pyruvate or citrate, and therefore that OXPHOS is sufficient to activate IFNγ responses. We used the inhibitor of pyruvate transport simply to show that the commercially-purchased basal media was properly formulated and the cells were dependent on the added pyruvate. These data are clearly not essential for the conclusions drawn, and their inclusion produced confusion. Thus, to ensure clarity we removed the data using the inhibitor which will not impact any conclusions.

7. The authors could consider placing their findings in the context of the broader literature. (As just one example, Ivashkiv Nat Imm 2015 described a role for mTORC1 and metabolism in IFNγ-mediated transcriptional and translational regulation in macrophages.) This would increase the impact of their findings.

As suggested, we have expanded our discussion of mTORC1 and IFNγ-mediated responses in the context of the Ivanshkiv 2015 manuscript to place our work in the broader context of IFNγ signaling regulation. This additional discussion and citations can be found in the introduction and discussion.

Reviewer #1:In this manuscript, Olive and colleagues used a genetic screen to identify mitochondrial Complex I as a regulator of IFNγ-mediated gene expression in macrophages. They attribute this role of CI to effects on the activity of the JAK-STAT pathway downstream of the IFNγ receptor. They show that T cell stimulatory activity is a consequence of CI activity.There are some questions regarding the authors' proposed mechanism for how CI activity influences IFN-mediated gene expression in macrophages:1. Lines 204-205: The authors find that sgRNAs targeting other complexes of the ETC, including CIII and CIV, had no effect on the ability of IFNγ to stimulate expression of cell surface markers. How do the authors interpret these findings, since CI does not work in isolation in the ETC and is rather dependent on CIII and CIV activity?

This is addressed in Point #1 of the essential revisions section.

2. How does IFNγ stimulation affect ETC activity, for example as assessed by Seahorse?

This is addressed in Point #2 of the essential revisions section.

3. Figure 6D: Not clear whether similar exposures were used in different panels. Would be better to load samples in the same gel so that the same exposure can be used and a direct comparison between conditions can be made.

This is addressed in Point #4 of the essential revisions section.

4. Figure 6D: Does acute treatment with rotenone have similar effects in reducing JAK-STAT signaling as knockdown of CI subunits? If not, then stable, long-term knockdown of CI subunits may have some effect independent of respiration in influencing JAK-STAT signaling (for example, on expression of some component of the JAK-STAT pathway). This interpretation could also explain why knockdown of other components of the ETC do not have similar effects to CI. Rotenone treatment could be tried and if the data are different from knockdown of CI subunits, then related data in the study could be re-interpreted and conclusions modified.

This is addressed in Point #5 of the essential revisions section.

5. The authors propose that "CI-dependent regeneration of NAD is the most likely regulator of IFNγ signaling". This conclusion is based on their observations that oligomycin but not CCCP has a similar effect to rotenone, but many other interpretations are possible. To suggest NAD regeneration as being causal seems a bit of stretch unless some supporting data is provided. For example, does addition of NAD precursors (e.g., NAM) rescue the defects in the CI subunit knockdown?

We agree that the mechanism linking CI and IFNg signaling remains unclear. The reasoning that lead us to favor NAD+ regeneration is laid out in a very circumspect manner and restricted to the discussion section. After re-reading this text, we do not feel that our result are over-interpreted. Instead, we used the discussion section to revisit the data and speculate on possible mechanisms. Upon reviewing this text we hope the reviewer will agree that this is an appropriate use of the discussion section.

Other concerns:1. The authors do not mention which macrophages (primary, cell line) they use for their screen.

This has been clarified in the main text.

Reviewer #2:– In figure 1E the authors suggest the importance of TNF receptor in regulating CD40 expression. Did the authors measure TNF production in the supernatant of their cell line treated with IFNγ? Based on the literature and on Figure 5C, there should be no TNF in the culture of IFNγ-treated macrophages. If so, how is it possible that the TNFR is involved? The authors should at least try to block TNF and/or TNFR with blocking antibodies to assess if suboptimal doses of TNF are playing any role.

While this issue is somewhat tangential to the major conclusions of the study and this comment was not listed as an “essential revision”, we agree that it is important to clearly state the limitations of our work. To be clear and transparent, the revised manuscript now states, “While these results do not define the full TNF-related signaling pathway, they are consistent with the specific association between TNF receptor expression and CD40 induction.”

– Throughout the paper: the authors often "measure" respiration and/or OXPHOS via measuring ATP levels (eg. Figure 3B and many others). The paper would highly benefit from the use of a seahorse to determine how the treatments made throughout the manuscript affect oxygen consumption and/or the extracellular acidification rate. Using ad hoc methods, it is possible not only to specifically measure OXPHOS and glycolysis, but also to ask important questions when substrates, such as pyruvate or citrate, are used. Most of the conclusions driven on the relevance of either respiration or OXPHOS, as well as ATP production sources, are very indirect and a seahorse would be required to really prove that the conclusions made are correct.

This is addressed in Point #2 of the essential revisions section. In the revised manuscript, we characterize the effects of IFNγ treatment by Seahorse, and only rely on ATP measurements to verify expected the metabolic deficiencies of engineered cell lines. We hope the reviewer agrees that the manuscript is improved and our conclusions are adequately supported.

– The introduction of the cytokine measurement in figure 4 and 5 is indeed important but quite misleading. No real comparison with the findings published by Ivashkiv's group in Nature Immunology in 2015 is made. This paper looks at the involvement of mTORC1, metabolism and other kinases in the transcriptional and translational regulation driven by IFNγ in macrophages. mTORC1 is a key regulator of glycolysis, and IFNγ is shown to inhibit mTORC1, if so, one would expect respiration and/or other metabolic changes may occur and may be somewhat linked to complex I. Also, the fact that in the absence/block of complex I, cytokines in cells treated with TLR2 agonists are boosted is very confounding, and it makes hard to interpret data when complex I is blocked and IFNγ is also administered together with TLR2 stimulation. The authors should either better explain these phenotypes, or only focus on signals that do not involve TLR2 stimulation, leaving these data for a future work.

This comment refers to the human macrophage studies in Figure 5B-C. We agree that these data are not mechanistically enlightening. However, in the larger context of macrophage function, we expect that readers will be interested in the effect of OXPHOS on cytokine production during combined stimulation with IFNγ and Pam3CSK4. If the reviewer finds the inclusion of these data counterproductive, we will gladly omit them. However, we prefer to include these observations, as they will answer an obvious question in many readers minds.

– In Figure 3H a key control is missing. What about survival of the cells when the import of the only energy substrate is blocked?

This is addressed in Point #6 of the essential revisions section.

– The experiment with CCCP suggests that Complex I may regulate macrophage functions independently of OXPHOS. This is maybe the most exciting point of the paper, and it is partially addressed in figure 6D where phosphorylation of JAKs and STATs is analyzed. How do we go from Complex I to the most proximal signaling components of the IFNGR?

That’s a great question. Our thoughts on this mechanism are outlined in the discussion.